



# Wetting and drying trends in the Land-Atmosphere Reservoir of large basins around the world

Juan F. Salazar[1], Ruben D. Molina[1], Jorge I. Zuluaga[2], and Jesus D. Gomez-Velez[3]

[1]GIGA, Escuela Ambiental, Facultad de Ingeniería, Universidad de Antioquia, Calle 70 No. 52-21, Medellín, Colombia.
[2]SEAP/FACom, Instituto de Física, Facultad de Ciencias Exactas y Naturales, Universidad de Antioquia, Calle 70 No. 52-21, Medellín, Colombia.
[3]Environmental Sciences Division & Climate Change Science Institute, Oak Ridge National Laboratory, 1 Bethel Valley Road, Oak Ridge, TN, 37830, USA

**Correspondence:** Juan F. Salazar (juan.salazar@udea.edu.co)

**Abstract.** Global change is altering hydrologic regimes worldwide, including large basins that play a central role in the sustainability of human societies and ecosystems. The basin water budget is a fundamental framework for understanding these basins' sensitivity and future dynamics under changing forcings. In this budget, studies often treat atmospheric processes as external to the basin and assume that atmosphere-related water storage changes are negligible in the long term. These assumptions are

potentially misleading in large basins with strong land-atmosphere feedbacks, including terrestrial moisture recycling, which is critical for global water distribution. Here we introduce the Land-Atmosphere Reservoir (LAR) concept to include atmospheric processes as a critical component of the basin water budget and use it to study long-term changes in the water storage of some of the world's largest basins. Our results show significant LAR water storage trends over the last four decades, with a marked latitudinal contrast: while tropical basins have been accumulating water, temperate basins have been drying. If continued, these

trends will disrupt the discharge regime and compromise the sustainability of these basins with widespread impacts.

## 1 Introduction: the Land-Atmosphere Reservoir

River basins are complex systems comprising physical, biological, and social components and a basic unit for studying the water cycle on land and implementing management and governance strategies (Cohen and Davidson, 2011). The sustainability of terrestrial ecosystems and human societies will depend on how river basins respond under the influence of global change

(Vörösmarty et al., 2010; Kuil et al., 2016; Mekonnen and Hoekstra, 2016; Best, 2019), including alterations due to climate change (Palmer et al., 2008), land use/land cover (LULC) change (Posada-Marín and Salazar, 2022), and other anthropogenic



stresses (Best, 2019). River discharge at the basin outlet is an integrated response resulting from the basin's water budget and, therefore, depending on the basin's properties and internal processes affecting terrestrial water fluxes and storage. Previous studies have identified changes in these fluxes and storage worldwide, including trends in precipitation (Lausier and Jain, 2018), river discharge (Barichivich et al., 2018; Li et al., 2020), terrestrial water storage (TWS) (Scanlon et al., 2018) and, generally, different components of the basin's water budget (Pan et al., 2012; Wang-Erlandsson et al., 2018; Zhang et al., 2019; Pabón-Caicedo et al., 2020). While critical for future sustainability, how and why the water budget of large basins is changing has yet to be fully understood (Pan et al., 2012; Jing et al., 2019; Posada-Marín and Salazar, 2022; Xiong et al., 2022). Here we first introduce the Land-Atmosphere Reservoir (LAR) concept for explicitly including the atmosphere in the basin water budget and then use this concept to show ongoing changes in some of the world's largest basins.

The most common approach to studying a basin's water budget, including theoretical, observational, and modeling studies, defines a control volume that includes the land and excludes the atmosphere (e.g., Pan et al., 2012; Kuil et al., 2016; Zhang et al., 2017, 2019; Posada-Marín and Salazar, 2022), i.e., a *land reservoir* (Fig. 1a,b). This control volume definition is a prevailing concept in catchment hydrology to study how basins respond to an external climatic input (Sivapalan, 2005; McDonnell et al., 2007) and for understanding human impacts on the water cycle (Abbott et al., 2019). From this perspective, precipitation is a flux that enters the basin from the exterior (it is regarded as an external forcing), and evapotranspiration represents a flux exiting the basin.

If defined as a land reservoir, the water budget equation for a basin,

$$R = P - E - \frac{\mathrm{d}S_L}{\mathrm{d}t}, \tag{1}$$

establishes that river discharge ($R$) depends on the difference between precipitation ($P$) and evapotranspiration ($E$), as well as on temporal changes in water storage within the land reservoir ($\mathrm{d}S_L/\mathrm{d}t$). The land reservoir (sometimes limited to a shallow soil layer) is widely used to define the control volume for computing a basin water budget in hydrological and land-surface models (e.g., Devia et al., 2015; Sood and Smakhtin, 2015; Blyth et al., 2021; Posada-Marín and Salazar, 2022). As a result, these models inherently assume that atmospheric processes exert external effects but do not make part of a basin's internal dynamics and water budget. This approach is the most widely used to simulate, for instance, the river discharge response to deforestation (Zhang et al., 2017; Posada-Marín and Salazar, 2022).

However, defining a basin system as a land reservoir may be misleading, especially for large basins with strong land-atmosphere feedbacks. For instance, let us consider the largest basin on Earth: the Amazon ($\approx 6$ million km$^2$, Fig. 1). Approximately 30% of rainfall that falls over the Amazon basin originates internally as evapotranspiration (Tuinenburg et al., 2020), mainly forest transpiration (Staal et al., 2018), resulting in the mechanism known as moisture (precipitation and evapotranspiration) recycling within the basin (Eltahir and Bras, 1994), i.e., local moisture recycling (LMR). Globally, 40% of the total rainfall falling over land comes from terrestrial evapotranspiration (van der Ent et al., 2010), and 57% of the rainfall over land returns to the atmosphere via evapotranspiration (Tuinenburg et al., 2020), meaning that moisture recycling from terrestrial sources plays a major role in distributing water over the land worldwide (te Wierik et al., 2021; Posada-Marín et al., 2023).

**Figure 1. The Land versus Land-Atmosphere Reservoirs. a,** Control volume and exchanges (precipitation, evapotranspiration, and discharge) for the land reservoir. See equation (1). **b,** Schematic representation of the land reservoir and exchanges in the Amazon basin, including the surface and land beneath it but excluding the atmospheric column. **c,** Control volume and exchanges (moisture convergence and discharge) for the LAR. See equation (2). **d,** Schematic representation of the LAR and exchanges in the Amazon basin, including the land reservoir and the atmospheric column above it. Moisture recycling occurs within the LAR. Imagery/Map data: ©2021 Google; Data SIO, NOAA, U.S. Navy, NGA, GEBCO; Landsat / Copernicus; IBCAO; INEGI. Basin polygon: GRDC.



Given its dependence on transpiration and, therefore, on the surface water budget and vegetation dynamics, LMR should
not be generally considered an external mechanism to the basin. In contrast, in cases such as large basins with strong land-
atmosphere feedbacks, this mechanism should be considered a crucial part of the system's internal dynamics, which plays a
role in regulating river discharge (Salazar et al., 2018) and is sensitive to anthropogenic effects such as LULC change (Ruiz-
Vásquez et al., 2020; te Wierik et al., 2021). To consider LMR and any other land-atmosphere interaction as part of a basin's
internal dynamics and their role in producing the river discharge regime, the control volume needs to be re-defined by including
the atmospheric column. The resulting land-atmosphere control volume is the *Land-Atmosphere Reservoir* or LAR (Fig. 1c,d),
i.e., the natural reservoir that receives water from the basin system's exterior, mainly through the atmosphere, and then stores
or releases it leading to the discharge regime.

The water budget equation for the LAR is

$$R = Q - \frac{\mathrm{d}(S_L + S_A)}{\mathrm{d}t}, \tag{2}$$

where river discharge ($R$) results from the difference between the net atmospheric convergence towards the basin system ($Q$)
and temporal changes in water storage within the LAR, including land ($S_L$) and atmospheric ($S_A$) components. In contrast to
the land reservoir, the water influx to the LAR is not precipitation but the atmospheric flux

$$Q = \oint_C \boldsymbol{\Theta} \cdot \mathrm{d}\boldsymbol{\ell}, \tag{3}$$

where $\boldsymbol{\Theta}$ is the vertically integrated atmospheric water flux and the integral is performed across the LAR's lateral contour
(more details in Section 2).

Equations (1) and (2) exclude a term representing the net convergence of groundwater. Unlike the atmospheric fields, the
global estimates of the groundwater flow field needed to estimate this underground convergence are limited. However, we do
not expect this term to significantly affect our results. Estimates of the continent-to-ocean groundwater flow show that this flow
is small relative to river discharge: $1\,\mathrm{km}^3\,\mathrm{yr}^{-1}$ compared to $1 \times 10^3\,\mathrm{km}^3\,\mathrm{yr}^{-1}$ in the Amazon basin, for example (more details
in Sections 2 and 3). Furthermore, groundwater fluxes in a large river basin contribute significantly to runoff and, therefore, are
largely accounted for in the outlet's river discharge.

A critical difference between the land reservoir and the LAR concepts is that in the latter, $P$ and $E$ are internal fluxes in the
basin system, allowing LMR to be a mechanism of the basin's internal dynamics that takes part in the basin water budget and,
therefore, in sustaining and regulating the discharge regime. A possible reason why the more traditional approach (the land
reservoir) excludes the atmosphere is that it has a much smaller water storage capacity than the land due to thermodynamic
constraints, suggesting the assumption that the atmosphere's role in the basin's internal dynamics, including changes in water
storage and regulation, is negligible. Although valid as a simplification in many cases (e.g., small watersheds where external
factors primarily impose precipitation), when applied to large basins, this assumption misses a fundamental feature of the
hydrological cycle: Despite its small storage capacity, the atmosphere has a vast capacity to transport water. Indeed, in the





global water budget, the inland transport of atmospheric moisture compensates for the offshore flow, including both surface water and groundwater (Trenberth et al., 2007). This transport capacity implies that a significant amount of water can be retained within the LAR by LMR (Fig. 1c), especially in large basins with high LMR rates.

Water stored within a basin's LAR through LMR involves not only atmospheric moisture but also the surface water that takes part in LMR, including every source of direct evaporation and transpiration in a basin. While from the land-reservoir perspective, evapotranspiration leaves the basin (Fig. 1a), the LAR perspective considers that significant amounts of transpired and evaporated water do not leave the basin but remain inside it through LMR (Fig. 1c). Hence, evapotranspiration is not necessarily a "loss" of water from the basin but can be a significant source of precipitation (e.g., see the "demand-side" and "supply-side" contrasting views discussed by Ellison et al., 2012).

Another difference between water storage dynamics in the LAR and the land reservoir is that the atmospheric processes and land-atmosphere interactions (occurring within the LAR but excluded from the land reservoir) are much more sensitive to climate change than, for instance, underground processes. These atmospheric processes include LMR as an essential component of the LAR's water storage and basins' internal dynamics and relate to the "green water" that is fundamental to the Earth system dynamics and is now extensively perturbed by human pressures at continental to planetary scales (Wang-Erlandsson et al., 2022).

Choosing between the land reservoir or LAR has important practical implications for modeling studies. Coe et al. (2009) compared results from models with land or land-atmosphere domains and showed that they produce contradictory results when investigating deforestation impacts on river discharge in some basins of South America. This contrast between results from models with land reservoir- or LAR-type domains is a general pattern across multiple studies (Posada-Marín and Salazar, 2022). A key reason is that models with land reservoir-type domains forced with measured precipitation do not "see" future changes in precipitation due to LULC change, including LULC impacts on LMR (or terrestrial moisture recycling in general).

Lastly, the LAR should not be confused with other established and related concepts such as moisture recycling (Eltahir and Bras, 1994) or the precipitationshed (Keys et al., 2012). While the LAR is a control volume, moisture recycling is a mechanism that can occur within it. The precipitationshed is not an Eulerian control volume such as the LAR, has different and more dynamic boundaries, excludes the land, and points to answer different questions (e.g., where does precipitable water come from?).

In the following sections, after describing data and methods, we use the LAR concept to study changes in the water budget of six of the largest basins on Earth, including tropical (the Amazon, Parana —most of its drainage is tropical—, and Congo) and temperate (the Mississippi, Ob, and Yenisei) river basins.

## 2 Data and methods

### 2.1 River discharge and its uncertainty

We obtained time series of monthly discharge, $R$ [m$^3$ s$^{-1}$], from the HYdro-geochemistry of the AMazonian Basin (HYBAM) observatory (Cochonneau et al., 2006) and the Global Runoff Data Centre (GRDC). We selected the following gauging stations





to maximize the drainage area and record length in each basin: Obidos for the Amazon River, Timbues for the Parana River,
Kinshasa for the Congo River, Vicksburg for the Mississippi River, Salekhard for the Ob River, and Igarka for the Yenisei
River. Supplementary Figures A1–A6 show the discharge time series used in our analysis.

HYBAM and GRDC do not report uncertainties in their discharge records. As a first-order approximation, we explored
relative errors in the discharge of 5% and 25%. The latter represents a conservative value for our uncertainty analysis. These
relative errors are consistent with the error estimates proposed by Syed et al. (2005), who assumed a relative error in the
observed Amazon and Mississippi discharge of 15%. Using these relative errors, we bound our estimates of changes in storage
and storage trends.

## 2.2 Moisture convergence and its uncertainty

We used 1979–2020 data from the ERA5 reanalysis (Hersbach et al., 2019) to estimate moisture convergence, $Q$ [$\mathrm{kg\,s^{-1}}$], for
each basin. Across a boundary $C$, $Q$ is defined by the contour integral shown in equation (3), where the vertically integrated
water flux, $\boldsymbol{\Theta}$ [$\mathrm{kg\,m^{-1}\,s^{-1}}$], is defined as

$$\boldsymbol{\Theta} = \frac{1}{g} \int\limits_0^{p_s} q\boldsymbol{v_h}\,\mathrm{d}p, \tag{4}$$

with $q$ [$\mathrm{g\,kg^{-1}}$] the specific humidity, $\boldsymbol{v_h}$ [$\mathrm{m\,s^{-1}}$] the horizontal wind field at each pressure level, $p$ [$\mathrm{kg\,m^{-1}\,s^{-2}}$] the total
air pressure, $p_s$ [$\mathrm{kg\,m^{-1}\,s^{-2}}$] the pressure at the Earth's surface, and $g$ [$\mathrm{m\,s^{-2}}$] the acceleration due to the Earth's gravity. $Q$
accounts for the vertically integrated atmospheric water fluxes in the solid, liquid, and vapor phases.

ERA5 provides monthly estimates of the eastward and northward components of the vertically integrated water fluxes within
a rectangular grid of $0.25° \times 0.25°$ resolution. We used this rectangular grid to rasterize each basin (Supplementary Fig. A7a)
and identify the grid edges defining its boundary (Supplementary Fig. A7b). From an implementation perspective, once the
boundary edges were defined, we differentiated them based on their orientation and whether the water flux was entering (inflow
edge) or leaving (outflow edge) the basin. For example, the edges oriented south-north were separated into inflow (Supplemen-
tary Fig. A7c) and outflow (Supplementary Fig. A7d) edges for eastward fluxes, the only flow component contributing to the
integral. Similarly, the edges oriented east-west were separated into inflow (Supplementary Fig. A7e) and outflow (Supple-
mentary Fig. A7f) edges for northward fluxes. As a convention, we assumed that inflow fluxes are positive and outflow fluxes
are negative. The discretized version of the contour integral defining $Q$ is estimated as the summation of the water fluxes
[$\mathrm{kg\,m^{-1}\,s^{-1}}$] crossing each edge multiplied by the edge's length [m] (Supplementary Fig. A8). Supplementary Figures A1–A6
show the resulting time series of $Q$.

ERA5 does not provide uncertainty estimates for $\boldsymbol{\Theta}$, or all the variables used for its calculation. Therefore, we cannot
simply propagate these variables' uncertainty through our approach to estimate $Q$. However, as part of their data assimilation
framework, ERA5 provides the ensemble spread at a coarser resolution ($0.5° \times 0.5°$) for a set of state variables (Hersbach et al.,
2020), including the vertically integrated water vapor divergence $D$ [$\mathrm{kg\,m^{-2}\,s^{-1}}$], which is a proxy for $Q$. This spread is not a





strict measure of uncertainty for the state variable estimates, as it ignores some important sources of error (e.g., systematic and correlated errors) (Asch et al., 2016), but provide a first-order approximation to bound our estimates of $Q$. More specifically, moisture convergence can be estimated as a function of water divergence with the divergence theorem by integrating $D$ over the basin area, i.e.,

$$Q = \int D \, \mathrm{d}S \tag{5}$$

with

$$D \equiv \frac{1}{g} \int\limits_0^{p_s} \boldsymbol{\nabla} \cdot (q\boldsymbol{v}_h) \, \mathrm{d}p. \tag{6}$$

Even though the spatial and temporal resolutions are different and the divergence only accounts for the vapor phase, the moisture convergence $Q$ computed with equations (3) and (5) show good agreement (Supplementary Figs. A9–A14 show the scatter plots). In other words, estimating the uncertainty of $Q$ by propagating the uncertainties of $D$ is reasonable. To do this,

we used linear propagation of uncertainties (Taylor, 1997) and the assumption of independent random errors to quantify the uncertainty in $Q$ as follows. First, a discretization of equation (5) allowed us to estimate moisture convergence, $\overline{Q_t}$, at a time $t$, as

$$\overline{Q_t} = \sum_{i=1}^{N_c} A_i D_{i,t}, \tag{7}$$

where $N_c$ is the number of grid cells within the basin, $A_i$ is the area of the $i$-th grid cell, and $D_{i,t}$ is the divergence value in

grid cell $i$ at time $t$. Under the previous assumptions, the error in $\overline{Q_t}$ is given by (Taylor, 1997)

$$\delta\overline{Q_t} = \sqrt{\sum_i^{N_c} (A_i \, \delta D_{i,t})^2} \tag{8}$$

where $\delta D_{i,t}$ is the error in the divergence $D_{i,t}$, assumed to equal the ensemble spread from ERA5. Finally, for a conservative estimate of the errors of $Q$, we assumed that the relative errors of $Q$ from equation (3) equal the relative errors of $Q$ from equation (5). That is

$$\delta Q(t) = \left( \frac{\delta\overline{Q_t}}{\overline{Q_t}} \right) Q(t). \tag{9}$$



### 2.3 Basin storage changes and its uncertainty

We used conservation of mass to estimate changes in the total LAR storage $(S_L + S_A)$. From the continuity equation (2), the LAR accumulates water, i.e., $\mathrm{d}(S_L+S_A)/\mathrm{d}t > 0$, when $Q > R$ and releases it, i.e., $\mathrm{d}(S_L+S_A)/\mathrm{d}t < 0$, when $Q < R$. Figure 2 uses the Amazon data to exemplify the schematic steps we followed to estimate these dynamics. First, we identified transitions between *accumulation and release periods*, corresponding to times where the $R$ and $Q$ time series intersect (vertical gray lines in Fig. 2). Supplementary Figures A1–A6 show these transitions for all basins.

Second, we calculated changes in water storage between transitions, $\Delta(S_L + S_A)$, by integrating the differences between $R$ and $Q$ over time, i.e., by solving

$$\Delta(S_L + S_A) = \int\limits_{\tau_1}^{\tau_2} [Q(t) - R(t)]\mathrm{d}t, \tag{10}$$

which represents accumulation within $(\Delta(S_L + S_A) > 0)$ or release from $(\Delta(S_L + S_A) < 0)$ the basin's LAR over the period between $\tau_1$ and $\tau_2$, with $\tau_1$ and $\tau_2$ being the onset and end of each accumulation or release period, respectively. Accumulation (green shaded bands in Figure 2a,b and Supplementary Figures A1–A6) occurs during prolonged periods (lasting from several days to months) when the atmospheric water converging into the LAR exceeds the river discharge (i.e., $Q > R$ so $\Delta(S_L + S_A) > 0$). Similarly, release (orange shaded bands in Figure 2a,b and Supplementary Figures A1–A6) occurs when discharge exceeds atmospheric water convergence (i.e., $Q < R$ so $\Delta(S_L + S_A) < 0$).

Third, we obtained the net accumulated or released volume from the onset of an accumulation period and the end of the next release period by adding consecutive volumes of accumulation and release (Fig. 2c). Finally, we obtained the long-term trends in the LAR's water storage by adding net accumulated or released volumes over time.

The accumulated storage shown in Figure 2d was calculated from data for $R$ and our estimates for $Q$. For convenience, the following discussion refers to these values as *nominal values* and includes bars in the variable names to emphasize their meaning (i.e., $\overline{R}$ and $\overline{Q}$). However, these values are uncertain and their uncertainty propagates through the storage calculations. We used a Monte Carlo analysis informed by the uncertainty metrics described for $R$ and $Q$ to estimate the uncertainty of our storage calculations and gain perspective regarding the robustness of our analyses and conclusions, as follows.

For each basin, we generated 1000 random realizations of the $R$ and $Q$ time series that preserve their correlation and error structure. Then, for each random realization of these fluxes, we identified the accumulation and release periods and estimated the corresponding storage change, rate of storage change, and accumulated storage. These new LAR storage metrics allowed us to bound the uncertainty in our estimates.

Individual realizations of the fluxes time series were generated by assuming that at any given time, $t$, the random variables $R_t$ and $Q_t$ are described by a multivariate normal distribution

$$[R_t, Q_t]^{\mathrm{T}} \sim \mathcal{N}([\overline{R}_t, \overline{Q}_t]^{\mathrm{T}}, \boldsymbol{\Sigma}_{\mathbf{RQ}}) \tag{11}$$

**Figure 2. Schematic steps to obtain accumulation and release periods and their metrics in the Amazon basin. a,** Time series of $R$ and $Q$ with accumulation and release periods, highlighting the period shown in the next panels. **b,** Accumulation and release periods. **c,** Net accumulated or released volume after two consecutive accumulation and release periods. **d,** Accumulated storage in the LAR after adding volumes in panel c.

with covariance matrix



$$\boldsymbol{\Sigma_{RQ}} = \begin{bmatrix} \delta R_t^2 & \rho_{RQ}\,\delta R_t\,\delta Q_t \\ \rho_{RQ}\,\delta R_t\,\delta Q_t & \delta Q_t^2 \end{bmatrix}, \tag{12}$$

where, for any time $t$, $\overline{R}_t$ and $\overline{Q}_t$ are the nominal discharge and moisture convergence values and $\delta R_t$ and $\delta Q_t$ are their absolute errors. Recall the two scenarios of relative errors in discharge we considered: (i) 5% or $\delta R_t = 0.05\,\overline{R}_t$ and (ii) 25% or $\delta R_t = 0.25\,\overline{R}_t$. Lastly, $\rho_{RQ}$ is the Pearson's correlation coefficient, defined as

$$\rho_{RQ} = \frac{1}{\sigma_R \sigma_Q} \sum_t (\overline{R}_t - \langle R \rangle)(\overline{Q}_t - \langle Q \rangle), \tag{13}$$

where $\langle R \rangle$ and $\langle Q \rangle$ correspond to the time average of the nominal values for both quantities. The difference $\langle Q \rangle - \langle R \rangle$ corresponds to the average LAR storage change, a quantity estimated from the mean annual cycles of $Q$ and $R$ in each basin (Supplementary Figs. A15–A20).

## 2.4 Estimating the annual cycle

We calculated the annual cycles of $Q$ and $R$ for each basin (Supplementary Figs. A15–A20) by transforming the time series to the phase domain. The phase associated with each point in the time series was calculated as an iterative optimization process, where we started with an arbitrary initial time, $t_0$, and assumed a cycle duration, $T$ (we used the tropical year duration of 365.24 days as an initial guess). Then, we divided the signal into time windows with a duration of $T$ days. In the $n$-th time window, which is contained between $t_n = t_0 + nT$ and $t_{n+1} = t_0 + (n+1)T$, the value of the phase for each point in the series is computed as $\phi = (t - t_n)/T$. After folding the signal, we found the average (solid lines in Supplementary Figs. A15–A20) and the envelope (maximum and minimum value of the signal). For a given pair of the free parameters $t_0$ and $T$, we computed the area of the envelope as a measure of the *goodness-of-folding* (GoF), which minimizes seasonal variability. Finally, we minimized GoF with respect to $t_0$ and $T$ for each basin and plotted the resulting envelopes.

## 2.5 Constraining groundwater discharge to the ocean

Our conceptual framework assumes that net groundwater fluxes leaving (or entering) the LAR control volume are small compared with (atmospheric) moisture convergence and discharge. Given that most of the fluxes exiting the large basins likely discharge into the ocean as submarine groundwater discharge (SGD), we present a back-of-the-envelope estimate to support our assumption. First, reported values of SGD are sparse, given the complexity when estimating these fluxes with environmental tracers, modeling, or a combination of both. Here, to obtain an order-of-magnitude estimate, we used an analysis by Sawyer et al. (2016) where annual volumetric discharge per unit length of the coast was estimated for the contiguous United States. In their analysis, the upper limit of the SGD is of the order of $1 \times 10^3\,\mathrm{m^2\,yr^{-1}}$. As an example, the coast length of the projected Amazon basin is of the order of $1 \times 10^6\,\mathrm{m}$. With these two values, we estimate that a reasonable upper limit for the





groundwater flux leaving the Amazon's LAR control volume and discharging to the ocean is of the order of $1\,\mathrm{km}^3\,\mathrm{yr}^{-1}$. This
order of magnitude is consistent for all the basins.

## 3   Results and discussion

### 3.1   The LAR in some of the world's largest basins

Figure 3a shows periods of net accumulation (green bars) and release (orange bars) for the Amazon basin and the corresponding
change in water storage estimated with equation (10). Supplementary Figures A21–A25 show the same results but for the other
basins.

The alternation between accumulation and release periods reflects seasonality in the basin, which in the Amazon is char-
acterized by the occurrence of one wet and one dry season over a period that is close to a year (Supplementary Fig. A15).
Changes in this seasonality are expected under global change (Costa and Pires, 2010; Fu et al., 2013; Wright et al., 2017),
potentially altering the LAR's dynamics and, therefore, the discharge regime.

Accumulation and release periods and their corresponding storage changes are not mirrored images of each other. Every pair
of consecutive accumulation and release periods produce a net change in water storage (Fig. 3b) that, if imbalanced over time,
produces long-term trends of accumulation (Fig. 3c) or release. If accumulation and release periods were always balanced,
there would not be long-term trends.

We found significant trends indicating that water storage has changed over the recent decades in all the basins' LAR (Fig.
4), with a marked latitudinal contrast: water storage has been increasing in tropical basins and decreasing in temperate ones.
These trends result from the accumulated imbalance between the LAR's water influx ($Q$) and efflux ($R$) (equation (2)). The
initial storage value is uncertain, so these trends have to be interpreted as changes in water storage relative to this initial value,
similar to the interpretation of TWS in GRACE studies.

GRACE studies serve as a reference for contextualizing LAR's storage trends. Notice, however, that even though TWS and
LAR storage are related, they are state variables describing the dynamics of different control volumes. Therefore, temporal
trends in these state variables do not have to be the same for a given basin. In a global study using three different GRACE
products for the period 2002–2014, Scanlon et al. (2018) reported TWS trends in our study basins varying from $-5\,\mathrm{km}^3\,\mathrm{yr}^{-1}$
in the Ob basin to $44\,\mathrm{km}^3\,\mathrm{yr}^{-1}$ in the Amazon basin. Roughly, this is equivalent to $-200\,\mathrm{km}^3$ to $1760\,\mathrm{km}^3$ over 40 years,
which is about one order of magnitude less than changes in the LAR's water storage over 1980–2020 (Fig. 4). Our results
coincide with Scanlon et al. (2018) in that the Amazon and Parana basins have been accumulating water after 2002, but not
for the Congo basin, where TWS has been slightly increasing (Scanlon et al., 2018) while the LAR's water storage has been
decreasing (Fig. 4). GRACE data are available only after 2002, so in this comparison between GRACE and LAR results, we are
considering only the trends shown in Figure 4 after that year. In temperate basins, results coincide for the Ob basin (decreasing
trend) but not for the Yenisei basin (increasing TWS trend). Scanlon et al.'s results for the Mississippi are mixed: they found
positive and negative trends in different sub-basins. Discrepancies between different GRACE products are common for large
basins, can be highly contrasting (e.g., positive versus negative trends), and remain a matter of investigation (Jing et al., 2019).

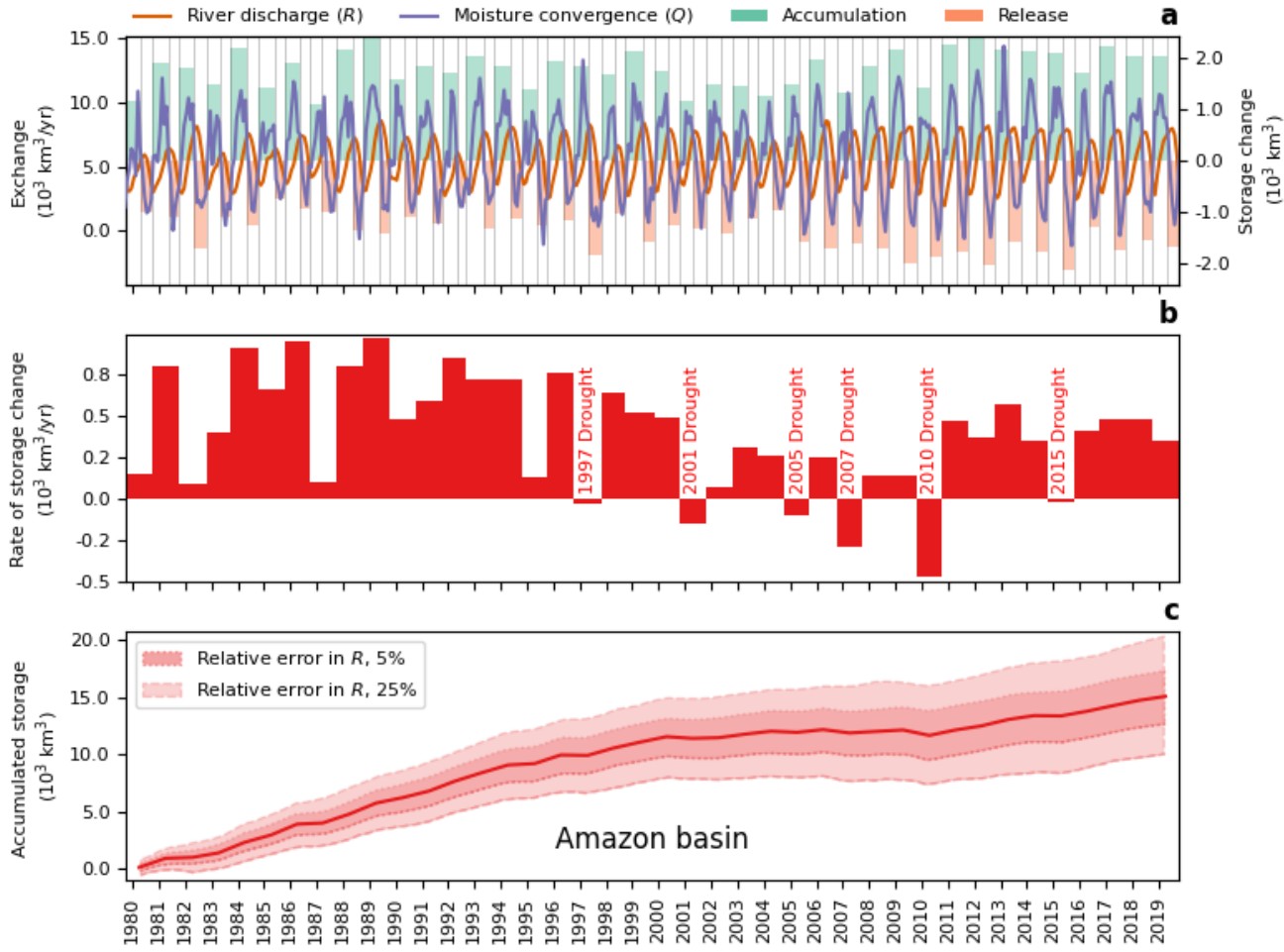

**Figure 3. LAR's dynamics in the Amazon river basin. a,** Monthly river discharge $R$ and net atmospheric convergence $Q$ (left axis). Green and orange bars show, respectively, the extent and volume (right axis) of accumulation and release periods. **b,** Net change in the LAR water storage after pairs of consecutive storage and release periods. **c,** Cumulative change in the LAR's water storage, including the corresponding errors in convergence and estimated discharge uncertainties (shadowed bands).

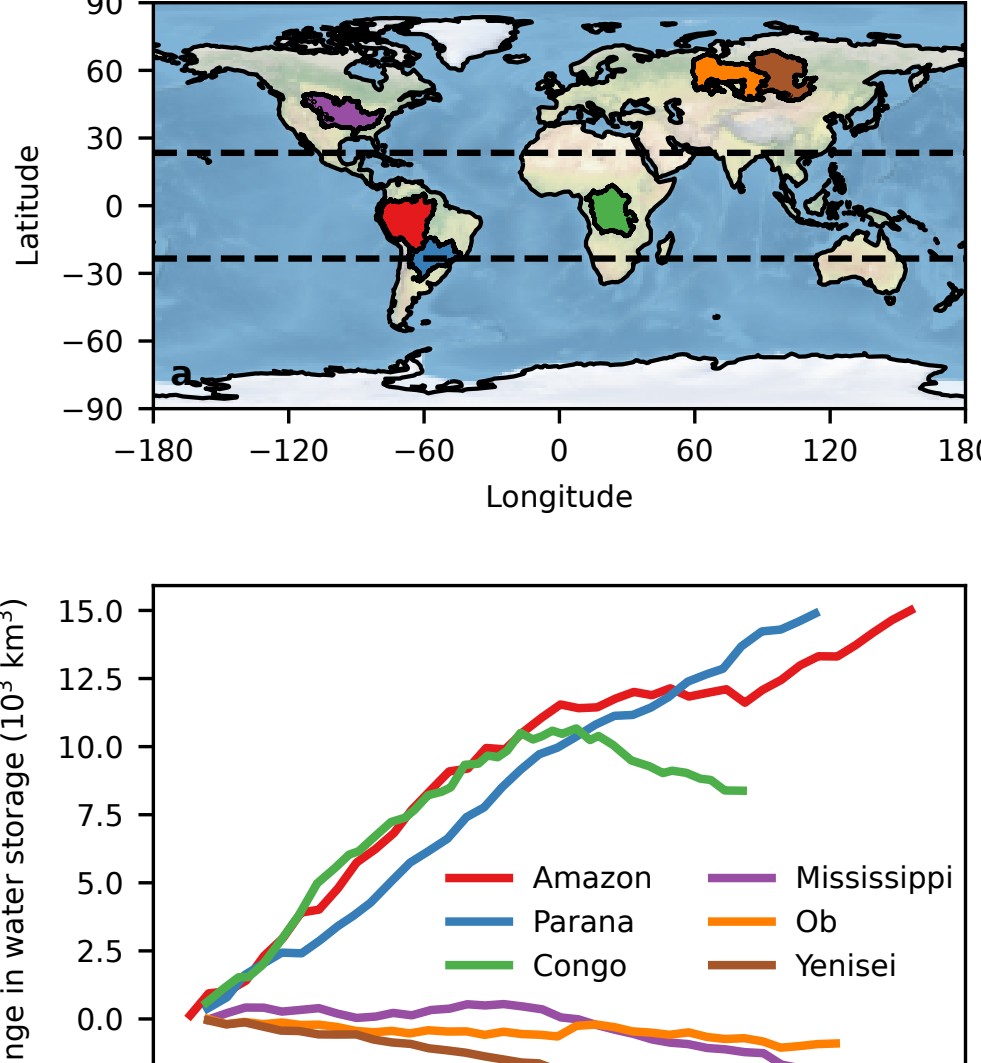

**Figure 4. Changing water storage in large basins' LAR. a,** Large basins in this study. **b,** Cumulative change in the LAR's water storage over time.





| Basin | Area $(\mathrm{km}^2)$ | $P$ $(\mathrm{mm\,yr^{-1}})$ | $P$ recycling rate (0–1) | Recycled volume $(\mathrm{km}^3\,\mathrm{yr}^{-1})$ | Average change in LAR's storage $(\mathrm{km}^3\,\mathrm{yr}^{-1})$ |
|---|---|---|---|---|---|
| Amazon | 4 690 963 | 2194 | 0.36 | 3706 | 390 |
| Congo | 3 634 880 | 1497 | 0.47 | 2558 | 296 |
| Parana | 2 527 003 | 1242 | 0.28 | 879 | 438 |
| Mississippi | 2 914 994 | 762 | 0.25 | 556 | −55 |
| Ob | 2 441 939 | 483 | 0.23 | 271 | −22 |
| Yenisei | 2 419 867 | 428 | 0.26 | 269 | −92 |

**Table 1.** Estimates of the recycled volume of water in each basin. Data sources: $P$ (Schneider et al., 2020) and $P$ recycling rate (Tuinenburg et al., 2020). LAR's averages correspond to Figures A15–A20.

Since TWS excludes the atmosphere (Wahr et al., 2004), it does not account for the water storage through LMR that depends on atmospheric water and dynamics. In contrast to the land reservoir and TWS measurements, the LAR's water storage inherently includes water circulation via LMR (Fig. 1). Annual recycled precipitation represents a water volume that is always

greater in magnitude than the average change in the LAR's water storage (Table 1), meaning that changes in LMR (e.g., driven by anthropogenic effects (te Wierik et al., 2021; Ruiz-Vásquez et al., 2020) or climate variability (Posada-Marín et al., 2023)) are potentially enough to explain the trends shown in Figure 4. The amount of water involved in LMR annually (recycled volume in Table 1) exceeds the average rates in TWS trends (Scanlon et al., 2018) by one to two orders of magnitude. Further, in the global water budget, the amount of atmospheric water entering the continents from the ocean ($\approx 40000$ km$^3$/yr) (Trenberth

et al., 2007) is two to three orders of magnitude greater than the average change in the LAR's storage (Table 1). These numbers show that, although seemingly counterintuitive, the idea that LMR can represent a significant part of a large basin's LAR water storage and contribute to explaining the found trends is plausible. Notice that this claim depends on the order of magnitude of the recycling ratio, which does not generally vary among studies (e.g., Dominguez et al., 2022), rather than on its "true" value that is uncertain and currently not directly measurable for vast regions.

**3.2    Confidence and uncertainty**

The "true" value of $Q$, $R$, and $d(S_A + S_L)/dt$ is unknown and difficult, if not impossible, to obtain with direct observations. We cannot measure $S_A$, $S_L$, or $Q$ directly and globally; even $R$ is hard to measure in vast rivers like the ones studied here. Further, TWS estimates can be contradictory among different GRACE products for reasons that remain unclear (Jing et al., 2019). The best we have are estimates based on different inherently uncertain techniques.

However, our uncertainty estimates indicate that the LAR trends are statistically robust (see bands in Fig. 3 and Supplementary Figs. A21–A25). The bands in panel c of these figures result from the uncertainty analysis explained in Section 2. The solid line represents the mean value of the accumulated storage and the bands the 5th and 95th percentiles of the Monte Carlo realizations for a relative error in $R$ of 5% and 25%. Hence, the width of these bands is a measure of the uncertainty in




our estimates of storage and how errors in the fluxes propagate through the analysis. Despite the uncertainties, the trends in
accumulated storage remain.

Besides the uncertainty estimates, we have several reasons to think that the LAR trends are plausible and indicative of important phenomena requiring attention. Every basin on Earth is under the influence of climate change, which, by definition, means trends and imbalance. The Earth's climate system has been imbalanced over the last centuries and will remain so over the coming decades, altering the water budgets globally (Xiong et al., 2022; Zaitchik et al., 2023).

Contrary to the widely-used assumption that changes in a basin water storage are negligible "in the long-term" (e.g., Poveda et al., 2007; Wang-Erlandsson et al., 2018; Hoek van Dijke et al., 2022), a growing body of literature shows that water fluxes entering and exiting the world's river basins are not necessarily balanced, so trends in water storage are not only plausible but likely. Wetting and drying trends are underway worldwide (Pan et al., 2012; Scanlon et al., 2018; Zhang et al., 2019; Pabón-Caicedo et al., 2020; Li et al., 2022; Xiong et al., 2022; Zaitchik et al., 2023). Examples include the study by Scanlon
et al. (2018) showing temporal changes in water storage inferred from GRACE data. The reduction of water storage due to permafrost thawing in large Siberian basins is consistent with LAR storage reductions in the Ob and Yenisei basins. A recent paper by Li et al. (2022) shows that basins draining from the Tibetan Plateau face drastic water availability reductions due to water storage losses, which implies long-term water budget imbalance in such basins.

The signal of droughts in the Amazon is notorious (Fig. 3): the basin's LAR has released water during documented droughts
in the last two decades, including the events of 1996–1997, 2001, 2004–2005, 2007, 2010, and 2015–2016 (Nepstad et al., 2004; Marengo et al., 2011; Tomasella et al., 2011; Jiménez-Muñoz et al., 2016; Tyukavina et al., 2017; Libonati et al., 2021). The largest release of water coincides with the record-breaking drought of 2010 (Marengo et al., 2011). This coincidence between LAR's release dynamics and severe droughts in the Amazon is unlikely a random error or systematic bias.

Also, the latitudinal contrast in the LAR trends is unlikely a random error or systematic bias. This contrast implies that
$\langle Q \rangle$ is larger than $\langle R \rangle$ in the South (tropical basins) and $\langle Q \rangle$ is smaller than $\langle R \rangle$ in the North (temperate basins), where the brackets represent long-term averages. If there was a systematic bias in our estimates based on ERA5 data, $Q$ should be consistently overestimated or underestimated. The latitudinal contrast suggests this would be the case only if ERA5 also has a latitude-dependent water budget bias, which would be an unknown bias requiring new evidence from future studies.

We also found temporal changes in the LAR trends. The most conspicuous case occurs in the Congo River basin, where
the slope changes sign (Fig. 4). If the trend does not reflect an actual phenomenon and ERA5 consistently overestimates or underestimates $Q$ for this basin, then there would not be a change in the trend slope. This change indicates that $\langle Q \rangle$ is larger than $\langle R \rangle$ during a period, and $\langle Q \rangle$ is smaller than $\langle R \rangle$ afterward.

Overall, our uncertainty analysis reinforces our main general conclusions about temporal changes in the LAR's water storage for some of the world's largest basins. The trends we found are plausible and statistically robust, providing fundamental insight
into the water storage dynamics constraining these big rivers' sustainability.





### 3.3 Regulation and sustainability

We use an artificial reservoir as an analogy to interpret our results. An artificial reservoir *regulates* river discharge either by mitigating floods through water accumulation or by enhancing low flows through water release, changing the river discharge regime. This reservoir's capacity to regulate discharge depends on the available volume to accumulate water during wet seasons and floods or to release previously stored water during dry seasons and droughts. Analogously, a basin's LAR can accumulate or release water leading to discharge regulation.


A basin's capacity to regulate river discharge depends on a complex and dynamic balance between accumulation and release processes (e.g., Fig. 3a) occurring within the whole LAR, not within the land reservoir alone. When a basin receives excessive water from the exterior (e. g., wet season) due to climate forcing (e. g., climate change or variability), discharge regulation manifests through temporal storage of water within the LAR, leading to discharge reduction, e. g., flood mitigation. Conversely, if the external water input is small (e. g., dry season or drought linked to reduced $Q$), regulating discharge (increasing low flow) requires the basin to release previously stored water.


The discovered trends (Fig. 4) affect these basins' regulation capacity, potentially compromising their river discharge regimes and sustainability. Since the regulation capacity requires available volume to store water during wet seasons (increased $Q$), a prolonged positive trend in the LAR's water accumulation (as we found in tropical basins) tends to reduce the LAR's capacity to store water. If continued, this trend will weaken the tropical basins' capacity to regulate river discharge by accumulating water. Such reduced storage capacity can combine with precipitation intensification due to climate change (Westra et al., 2013; Zhang et al., 2013) to weaken the tropical basins' capacity to mitigate (regulate) floods. We think that this regulation weakening in the LAR is a previously unknown mechanism behind the marked increase in very severe floods observed over recent decades in the Amazon (Marengo and Espinoza, 2016; Barichivich et al., 2018), related but not limited to a reduced storage capacity of the land reservoir (Reager and Famiglietti, 2009).



The trend reversal in the Congo basin (around the year 2000, the trend slope changes from positive to negative, Fig. 4) suggests the possibility of longer-scale transitions between accumulation and release periods, possibly leading to regulation patterns at the scale of centuries. The possibility of confirming this is limited by the length of available records. Regardless of the case, decadal trends and their impacts can strongly affect river discharge regimes and should be monitored.


The negative trend in the LAR's water storage reduces the temperate basins' capacity to enhance low flow by releasing previously stored water. Hence, if continued, these negative trends can combine with more extreme droughts due to climate change (Mann and Gleick, 2015) to weaken these basins' capacity to regulate low flows. Continuous storage reduction in the Ob and Yenisei rivers coincides with permafrost thawing, which is a driver of discharge increase in these Siberian basins, especially in winter (Wang et al., 2021) (see also the Supplementary Figs. A19 and A20). Analogously, our results suggest that the observed increase in the Mississippi river discharge (Shi et al., 2019) has occurred at the expense of storage reduction that is noticeable in the LAR (Fig. 4). Non-perennial rivers and streams are common in the Mississippi, Ob, and Yenisei basins (Messager et al., 2021) and will become more common if the LAR's drying trends continue.




## 4 Conclusions

We studied the water budget of six of the largest river basins on Earth (the Amazon, Parana, Congo, Mississippi, Ob, and Yenisei) through the lens of the LAR. The LAR is a control volume that explicitly includes land-atmosphere interactions, such as moisture recycling, as part of these basins' internal dynamics. This definition contrasts the more traditional perspective, which we described as the land reservoir, which considers the atmosphere external to river basins and precipitation as an external forcing.

Using observational and reanalysis data and the water budget equation for the LAR, we found trends in water storage within the studied basins' LAR, exhibiting a marked latitudinal contrast: while tropical basins are getting wetter, temperate basins are getting drier. These patterns result from long-term imbalances in which tropical basins have received more water through the atmosphere than they have released through river discharge. The opposite has occurred in temperate basins. As for our uncertainty analysis, these trends are robust.

If continued, the found trends may disrupt the basins' river discharge regimes. More specifically, sustained long-term increases in the water storage of the tropical basin's LAR (wetting trends) could reduce these basins' capacity to mitigate floods through water storage during wet seasons. Likewise, drying trends can reduce the temperate basins' capacity to sustain low flows by releasing previously stored water during dry seasons or droughts. The LAR provides a framework for monitoring and further investigating these changes, which are critical for the sustainability of human societies and ecosystems in the face of 365 climate change.

*Code and data availability.* The code used in this study will be available in a public repository before publication. All the data used in this study is publicly available. The datasets generated and/or analysed during the current study will be available in a public repository before publication.

*Author contributions.* All authors conceived the idea and methodological approach. J.F.S. wrote the first draft and all authors reviewed and 370 edited the manuscript. All authors implemented and ran the analyses.

*Competing interests.* None of the authors has any competing interests.

*Acknowledgements.* J.F.S. (lead investigator), R.D.M., and J.I.Z. were funded by the Colombian Ministry of Science, Technology and Innovation (MINCIENCIAS) through the SOS-Cuenca research program "SOStenibilidad de sistemas ecológicos y sociales en la CUENCA Magdalena-Cauca bajo escenarios de cambio climático y pérdida de bosques" (code 1115-852-70719) with funds from "Patrimonio Autónomo 375 Fondo Nacional de Financiamiento para la Ciencia, la Tecnología y la Innovación, Fondo Francisco José de Caldas". J.D.G-V. was funded

<br>




by the U.S. Department of Energy, Office of Science, Biological and Environmental Research. This work is a product of two programs: (i) Environmental System Science Program, as part of the Watershed Dynamics and Evolution (WADE) Science Focus Area (SFA) at ORNL and the IDEAS-Watersheds project, and (ii) Data Management Program, as part of the ExaSheds project. Additional support was provided by the National Science Foundation (awards EAR-1830172, OIA-2020814, and OIA-2312326).





**Appendix A:  Supplementary Figures**

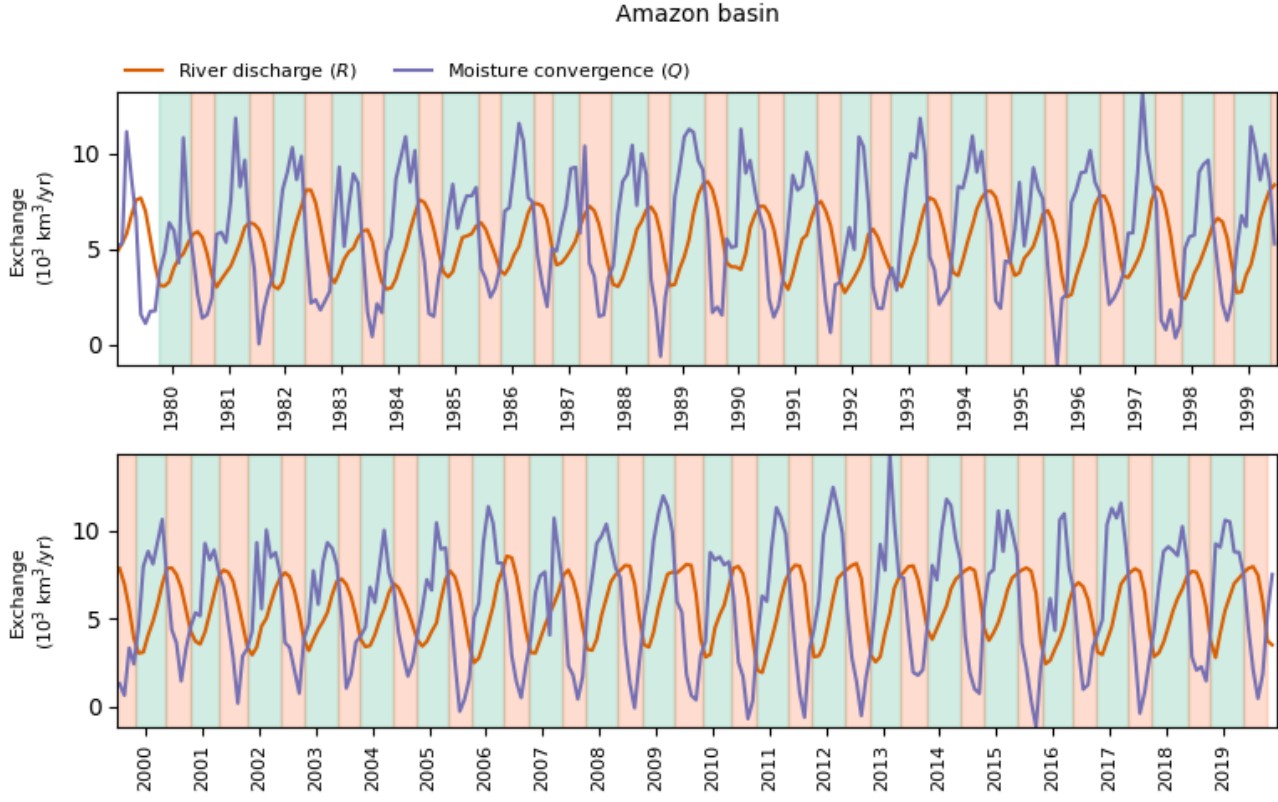

**Figure A1.** Identification of the onset and end of accumulation (green) and release (orange) periods in the Amazon basin.

**Figure A2.** Identification of the onset and end of accumulation (green) and release (orange) periods in the Parana basin.

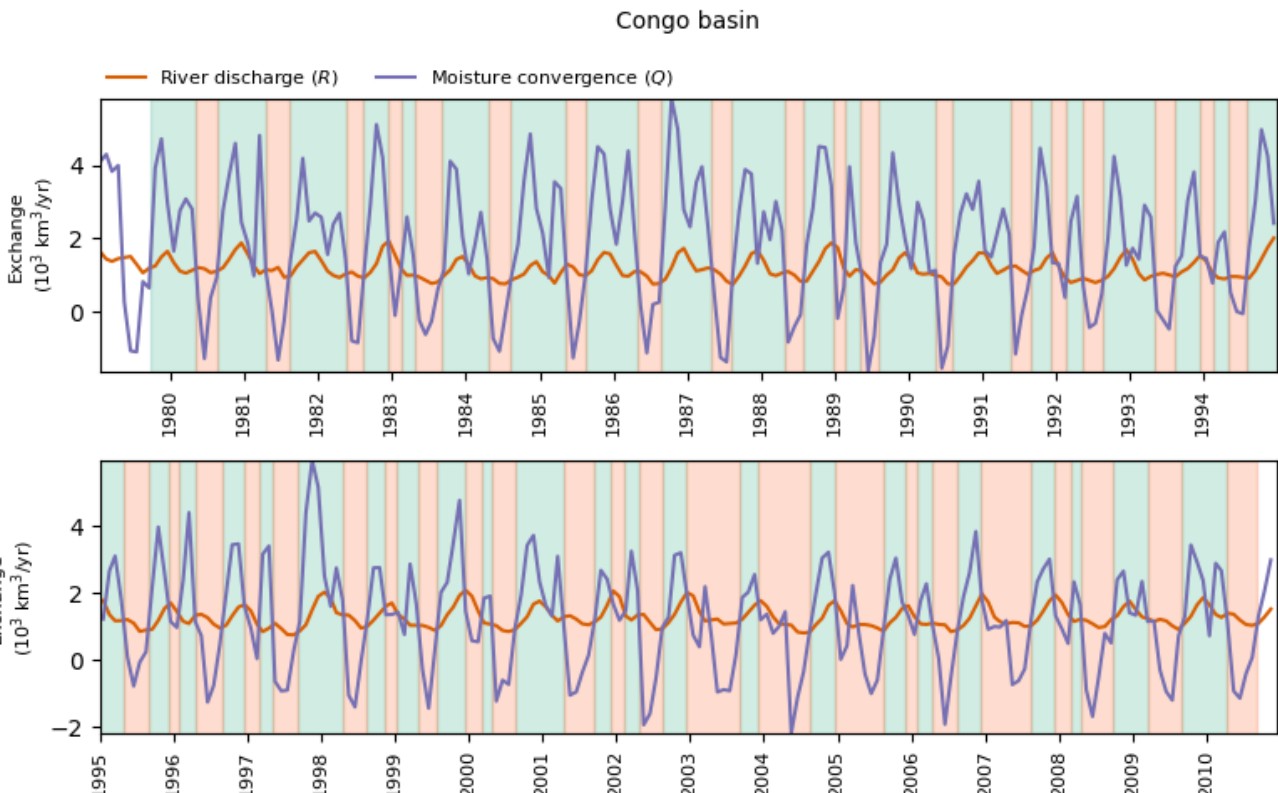

**Figure A3.** Identification of the onset and end of accumulation (green) and release (orange) periods in the Congo basin.



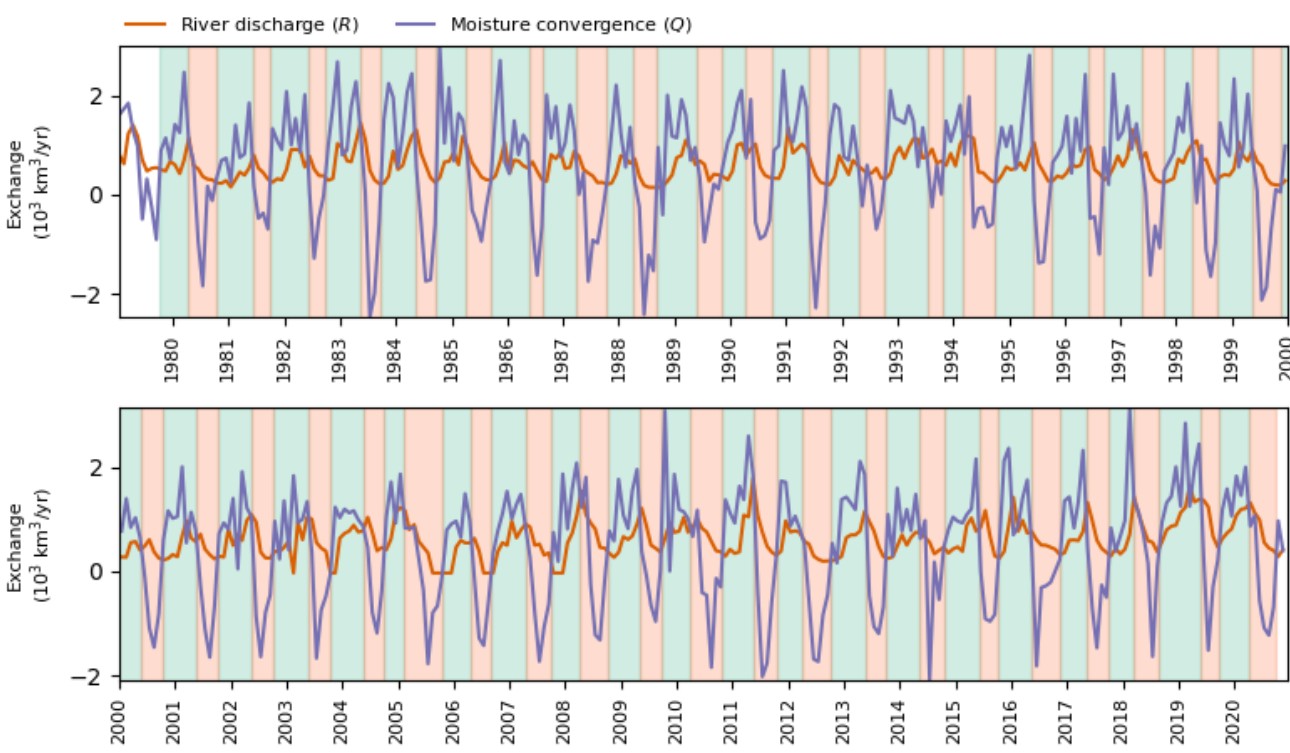

**Figure A4.** Identification of the onset and end of accumulation (green) and release (orange) periods in the Mississippi basin.



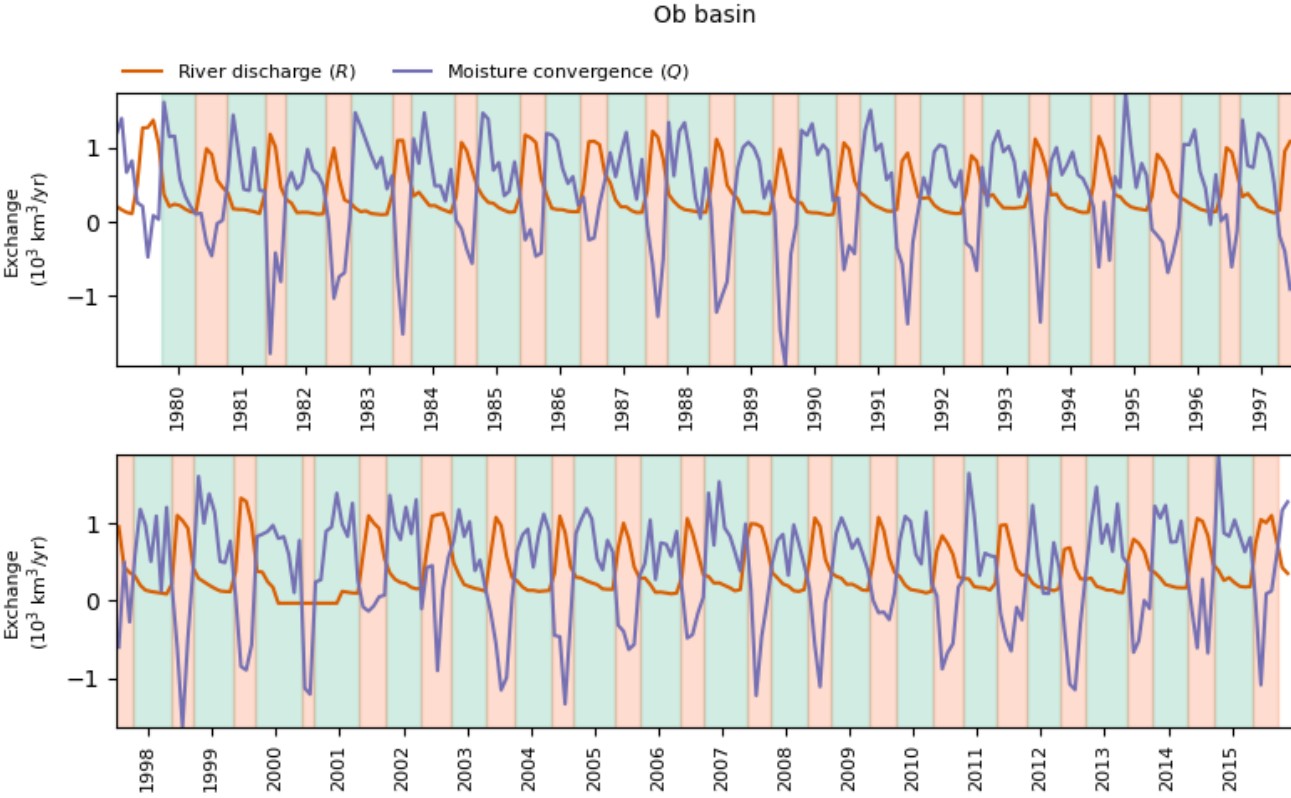

**Figure A5.** Identification of the onset and end of accumulation (green) and release (orange) periods in the Ob basin.



**Figure A6.** Identification of the onset and end of accumulation (green) and release (orange) periods in the Yenisei basin.

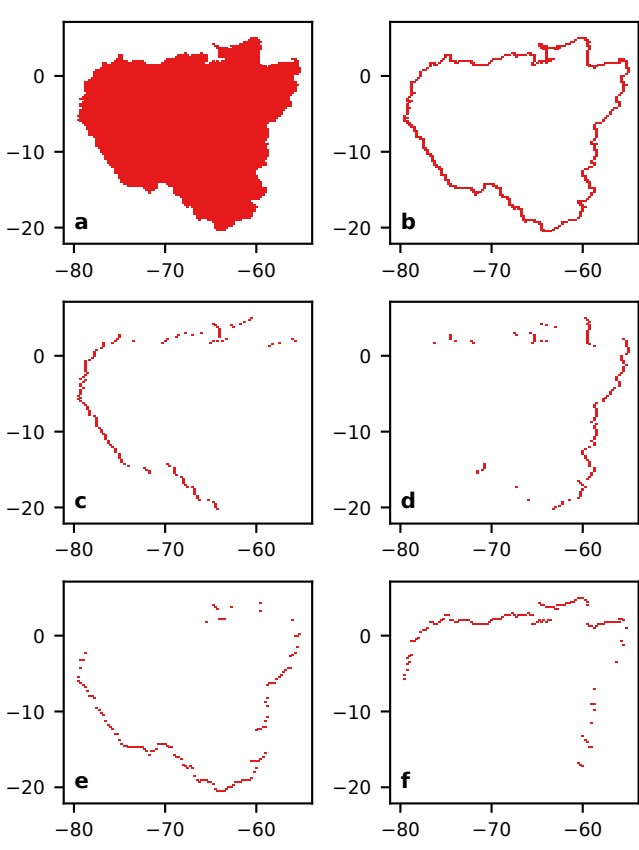

**Figure A7. Identification of the inflow and outflow edges used to compute moisture convergence in the Amazon basin. a,** Rasterization of the basin polygon with the ERA5 latitude-longitude rectangular grid. **b,** Identification of the basin contour edges. **c,** Inflow edges for eastward fluxes. **d,** Outflow edges for eastward fluxes. **e,** Inflow edges for northward fluxes. **f,** Outflow edges for northward fluxes.



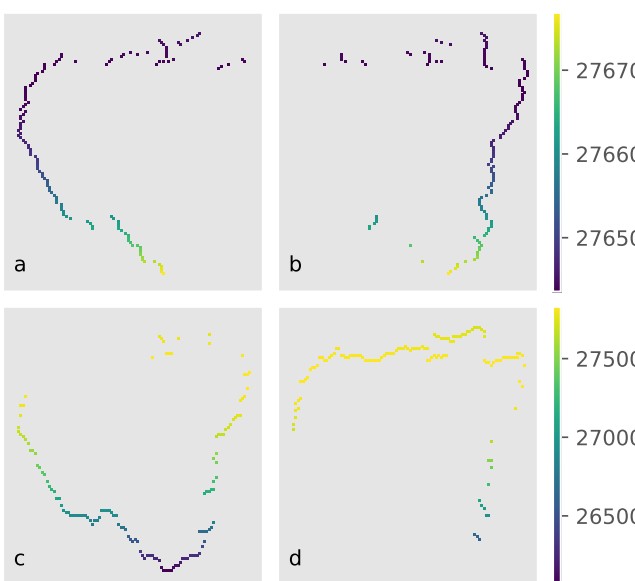

**Figure A8. Length** (m) **of the contour edges for the Amazon basin. a,** Inflow edges for eastward fluxes. **b,** Outflow edges for eastward fluxes. **c,** Inflow edges for northward fluxes. **d,** Outflow edges for northward fluxes.



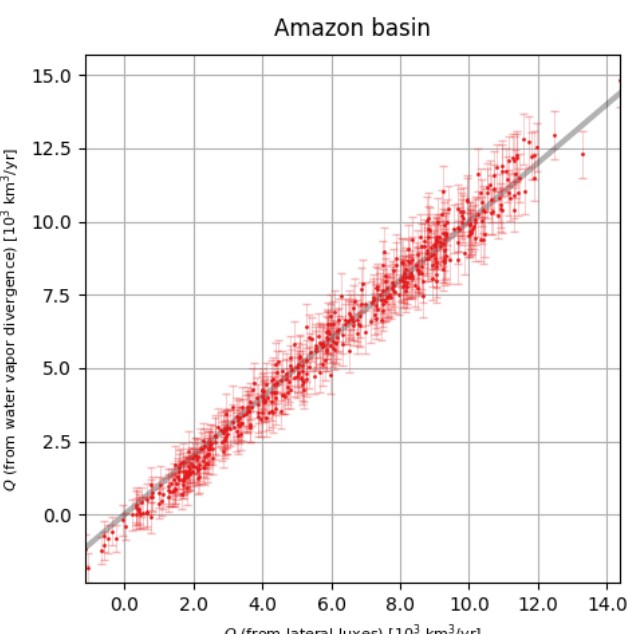

**Figure A9.** Comparison of moisture convergence estimated from the vertically integrated water flux (equation (3); x-axis) and vertical integral of the divergence of water vapor (equation (5); y-axis) for the Amazon basin. Each point corresponds to the monthly average of $Q(t)$ during the time span available in the ERA5 data products. Error bars are calculated with equation (8).



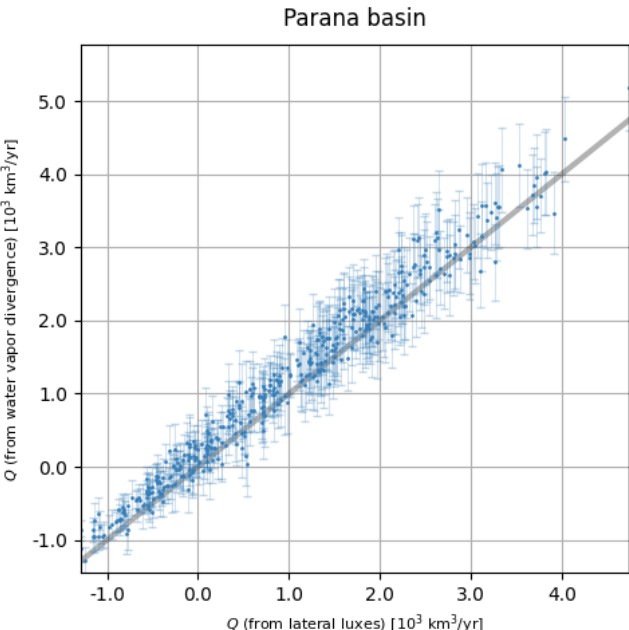

**Figure A10.** Comparison of moisture convergence estimated from the vertically integrated water flux (equation (3); x-axis) and vertical integral of the divergence of water vapor (equation (5); y-axis) for the Parana basin. Each point corresponds to the monthly average of $Q(t)$ during the time span available in the ERA5 data products. Error bars are calculated with equation (8).



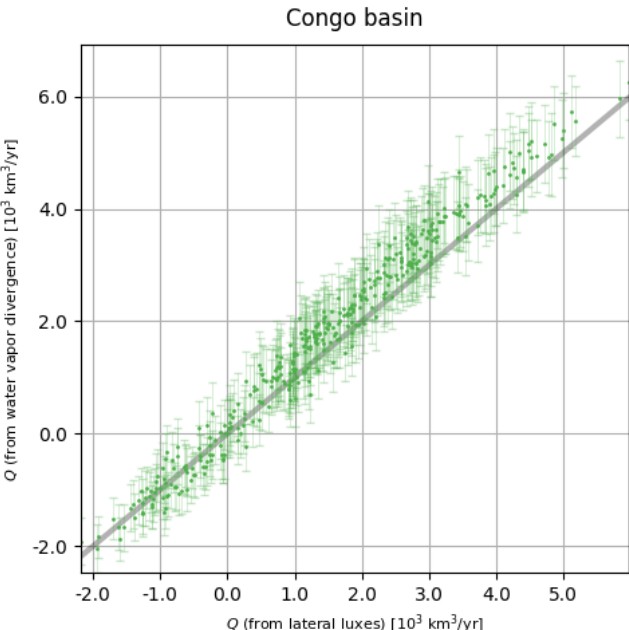

**Figure A11.** Comparison of moisture convergence estimated from the vertically integrated water flux (equation (3); x-axis) and vertical integral of the divergence of water vapor (equation (5); y-axis) for the Congo basin. Each point corresponds to the monthly average of $Q(t)$ during the time span available in the ERA5 data products. Error bars are calculated with equation (8).



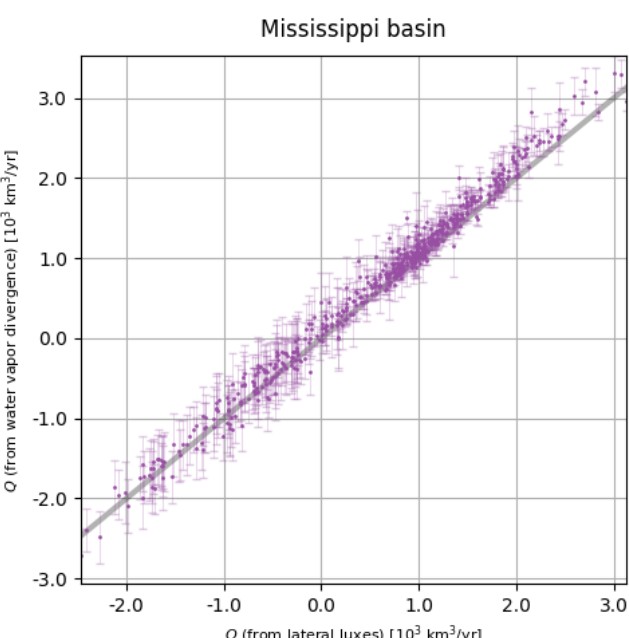

**Figure A12.** Comparison of moisture convergence estimated from the vertically integrated water flux (equation (3); x-axis) and vertical integral of the divergence of water vapor (equation (5); y-axis) for the Mississippi basin. Each point corresponds to the monthly average of $Q(t)$ during the time span available in the ERA5 data products. Error bars are calculated with equation (8).

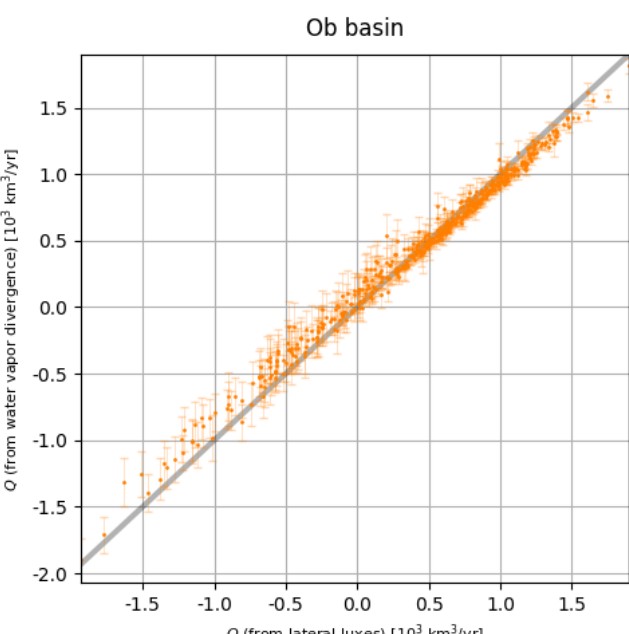

**Figure A13.** Comparison of moisture convergence estimated from the vertically integrated water flux (equation (3); x-axis) and vertical integral of the divergence of water vapor (equation (5); y-axis) for the Ob basin. Each point corresponds to the monthly average of $Q(t)$ during the time span available in the ERA5 data products. Error bars are calculated with equation (8).





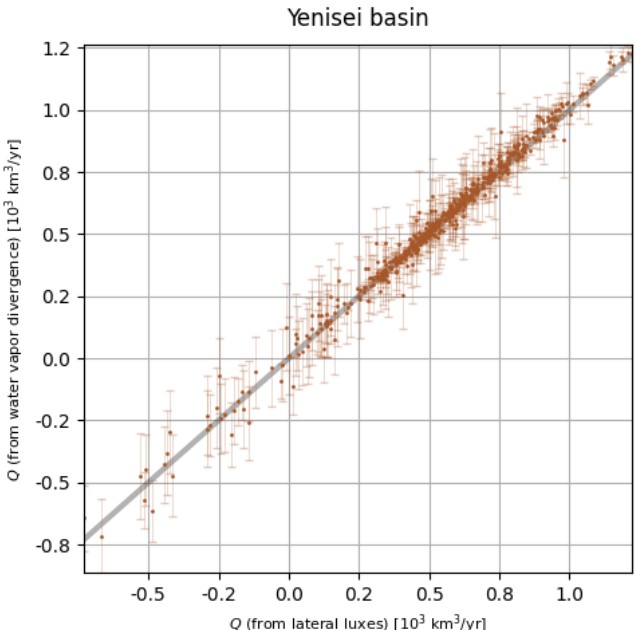

**Figure A14.** Comparison of moisture convergence estimated from the vertically integrated water flux (equation (3); x-axis) and vertical integral of the divergence of water vapor (equation (5); y-axis) for the Yenisei basin. Each point corresponds to the monthly average of $Q(t)$ during the time span available in the ERA5 data products. Error bars are calculated with equation (8).

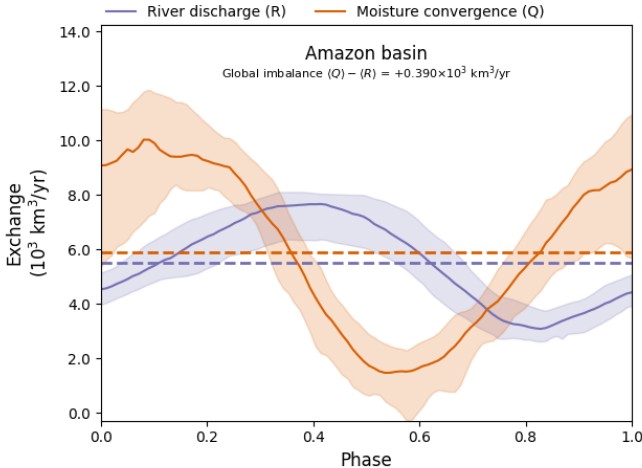

**Figure A15.** Annual cycle of LAR exchanges for the Amazon basin. Solid line corresponds to the seasonal average and shaded area to the corresponding envelope. Dashed lines show long-term average river discharge and moisture convergence.





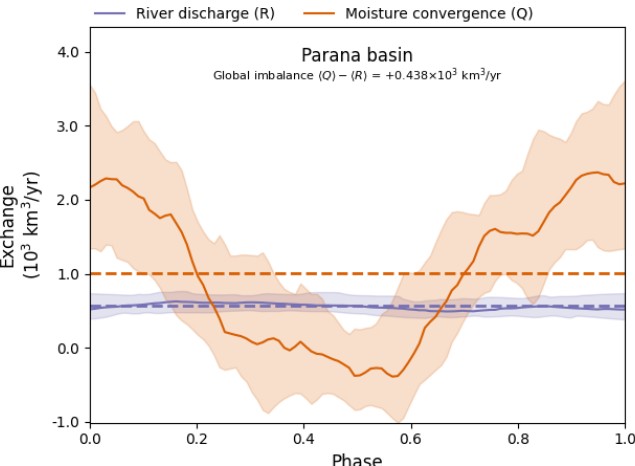

**Figure A16.** Annual cycle of LAR exchanges for the Parana basin. Solid line corresponds to the seasonal average and shaded area to the corresponding envelope. Dashed lines show long-term average river discharge and moisture convergence.

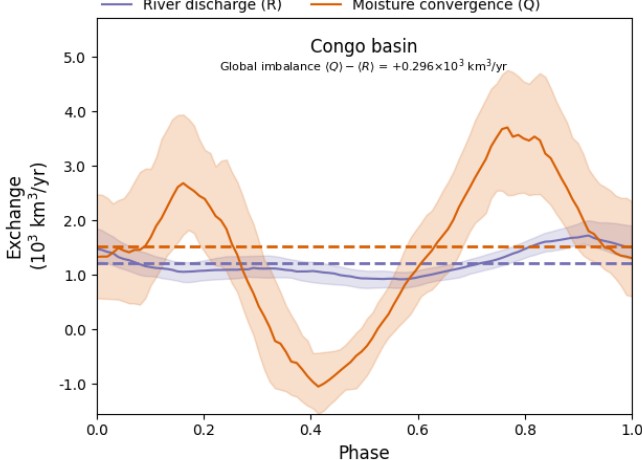

**Figure A17.** Annual cycle of LAR exchanges for the Congo basin. Solid line corresponds to the seasonal average and shaded area to the corresponding envelope. Dashed lines show long-term average river discharge and moisture convergence.





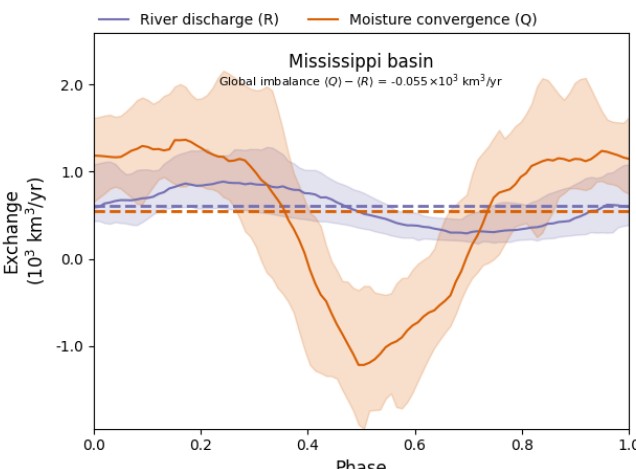

**Figure A18.** Annual cycle of LAR exchanges for the Mississippi basin. Solid line corresponds to the seasonal average and shaded area to the corresponding envelope. Dashed lines show long-term average river discharge and moisture convergence.

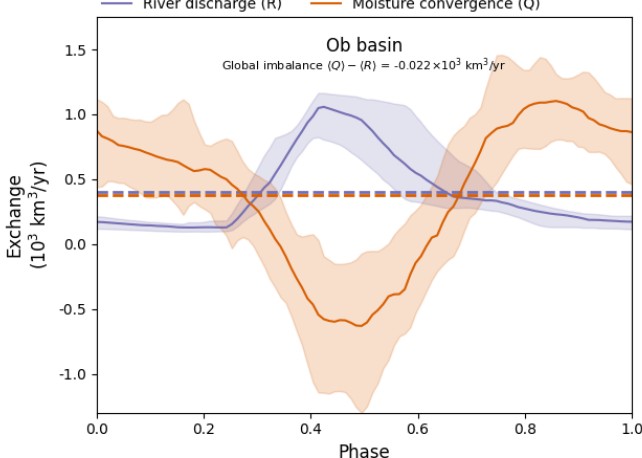

**Figure A19.** Annual cycle of LAR exchanges for the Ob basin. Solid line corresponds to the seasonal average and shaded area to the corresponding envelope. Dashed lines show long-term average river discharge and moisture convergence.



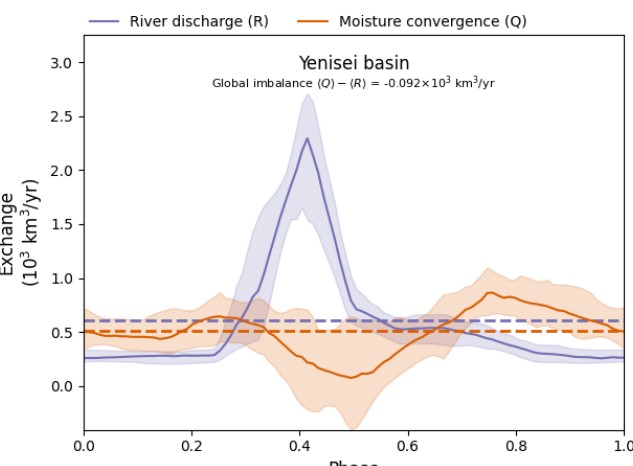

**Figure A20.** Annual cycle of LAR exchanges for the Yenisei basin. Solid line corresponds to the seasonal average and shaded area to the corresponding envelope. Dashed lines show long-term average river discharge and moisture convergence.

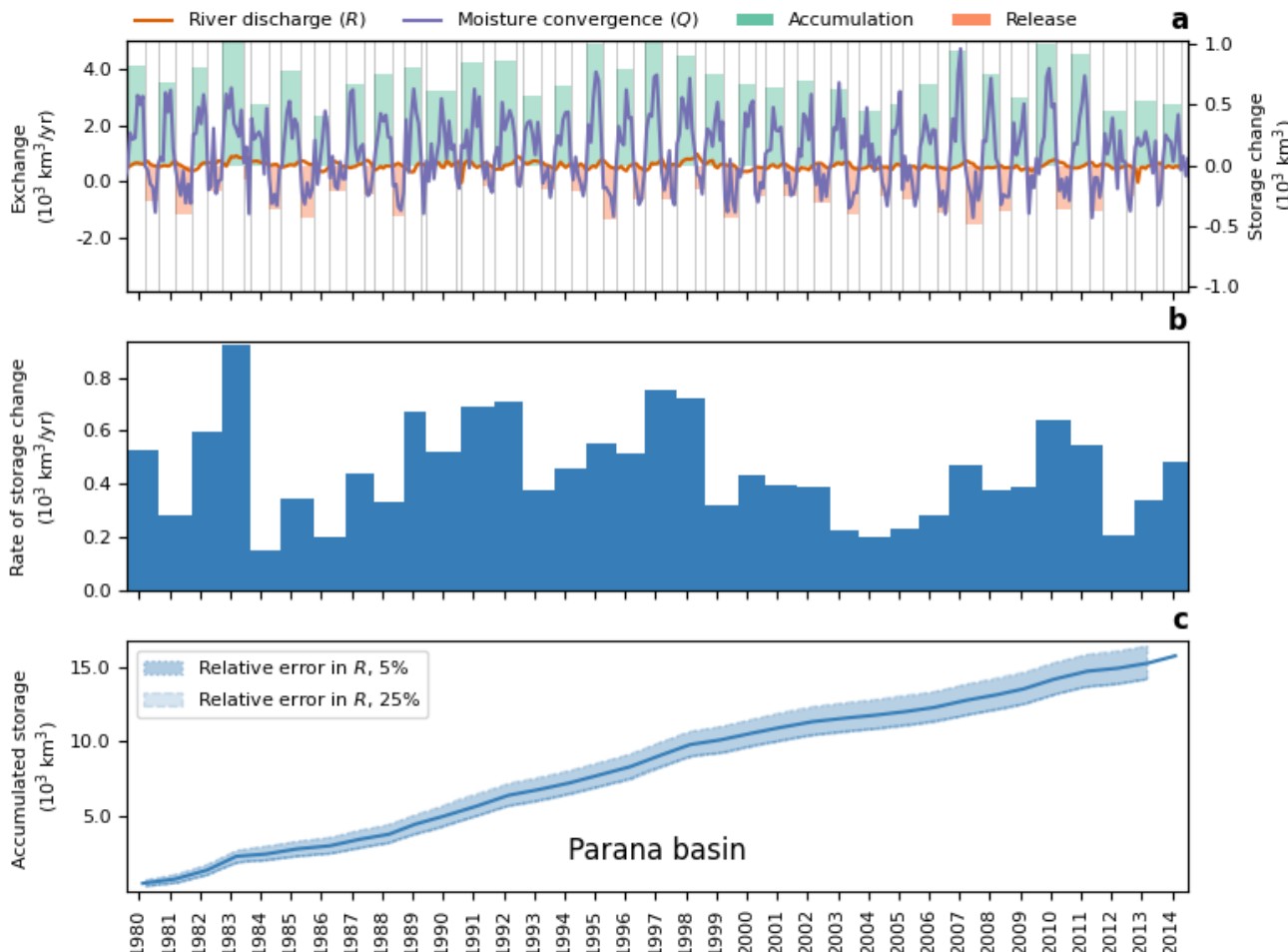

**Figure A21. LAR's dynamics in the Parana river basin. a,** Monthly river discharge $R$ and net atmospheric convergence $Q$ (left axis). Green and orange bars show, respectively, the extent and volume (right axis) of accumulation and release periods. **b,** Net change in the LAR water storage after pairs of consecutive storage and release periods. **c,** Cumulative change in the LAR's water storage, including the corresponding errors in convergence and estimated discharge uncertainties (shadowed bands). See Methods for more details.

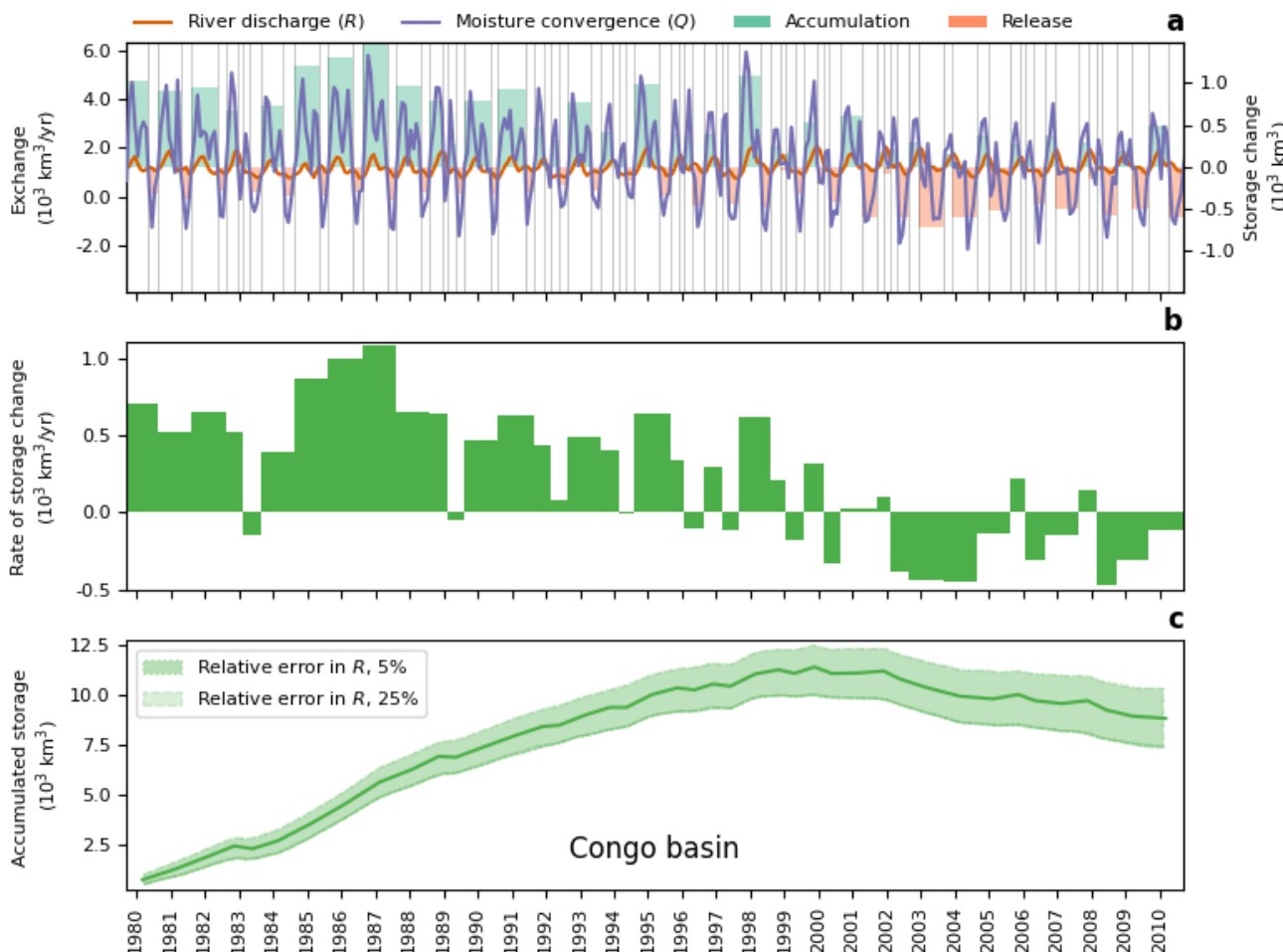

**Figure A22. LAR's dynamics in the Congo river basin. a,** Monthly river discharge $R$ and net atmospheric convergence $Q$ (left axis). Green and orange bars show, respectively, the extent and volume (right axis) of accumulation and release periods. **b,** Net change in the LAR water storage after pairs of consecutive storage and release periods. **c,** Cumulative change in the LAR's water storage, including the corresponding errors in convergence and estimated discharge uncertainties (shadowed bands). See Methods for more details.

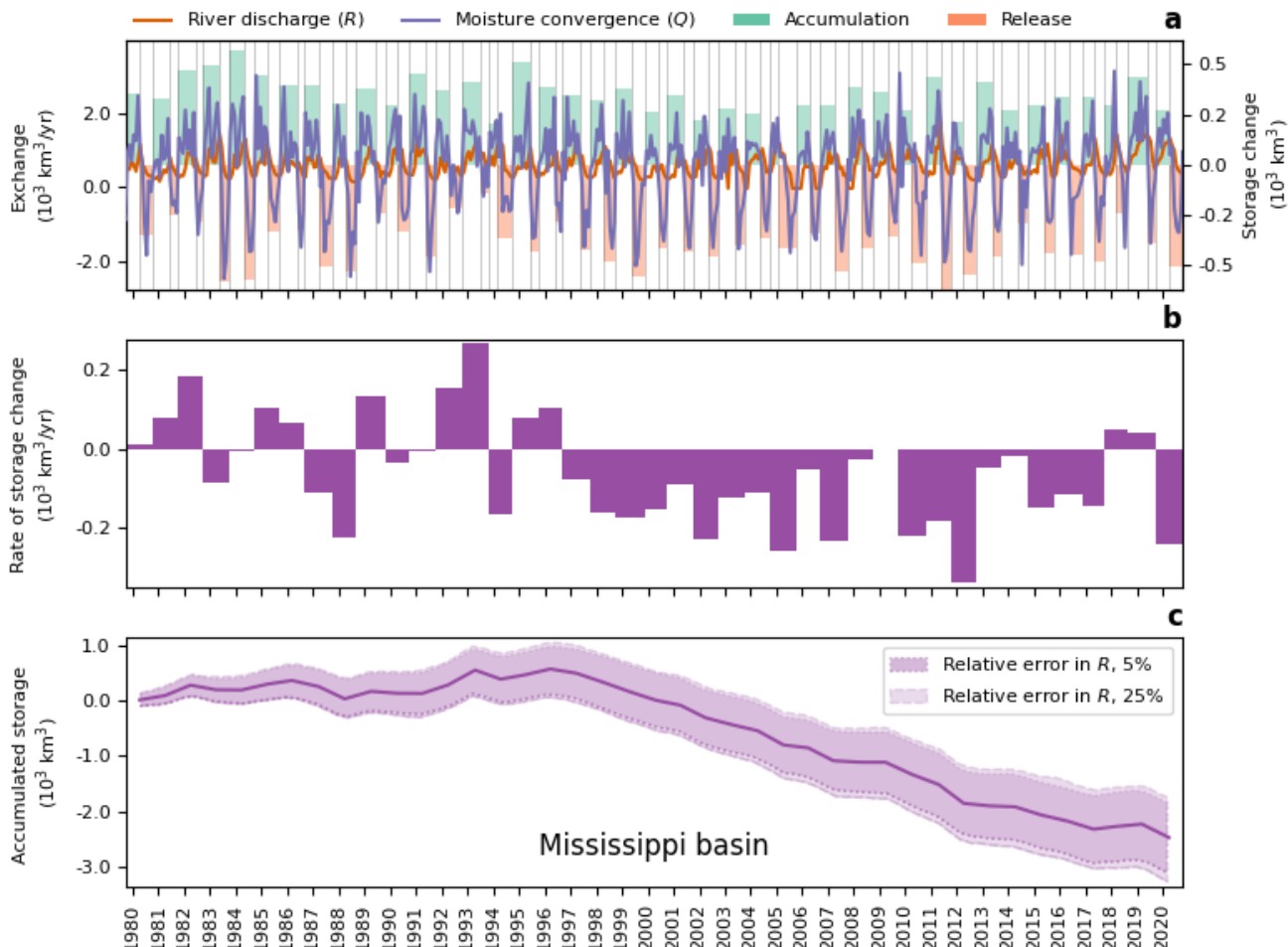

**Figure A23. LAR's dynamics in the Mississippi river basin. a,** Monthly river discharge $R$ and net atmospheric convergence $Q$ (left axis).

Green and orange bars show, respectively, the extent and volume (right axis) of accumulation and release periods. **b,** Net change in the

LAR water storage after pairs of consecutive storage and release periods. **c,** Cumulative change in the LAR's water storage, including the

corresponding errors in convergence and estimated discharge uncertainties (shadowed bands). See Methods for more details.

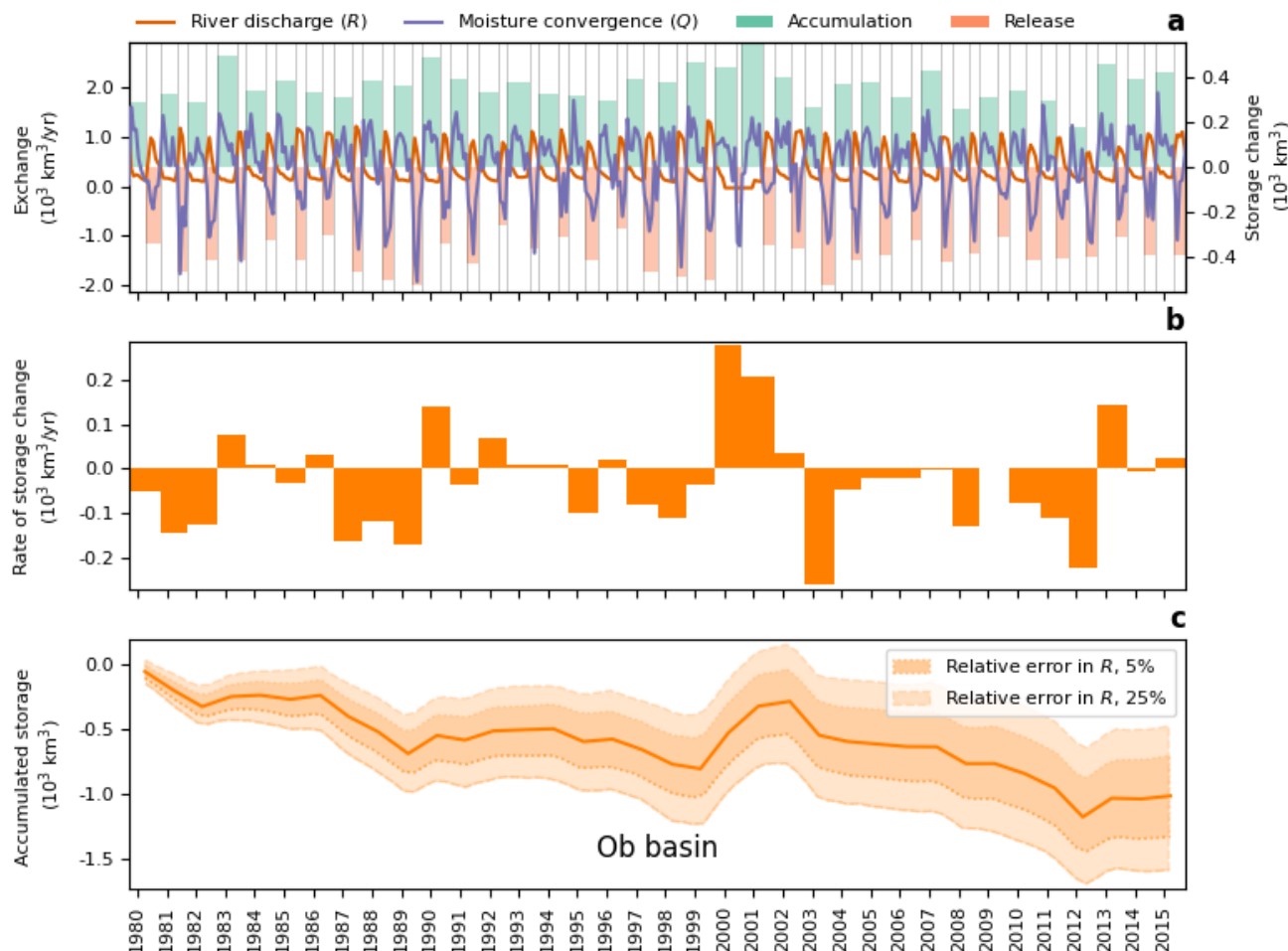

**Figure A24. LAR's dynamics in the Ob river basin. a,** Monthly river discharge $R$ and net atmospheric convergence $Q$ (left axis). Green and orange bars show, respectively, the extent and volume (right axis) of accumulation and release periods. **b,** Net change in the LAR water storage after pairs of consecutive storage and release periods. **c,** Cumulative change in the LAR's water storage, including the corresponding errors in convergence and estimated discharge uncertainties (shadowed bands). See Methods for more details.

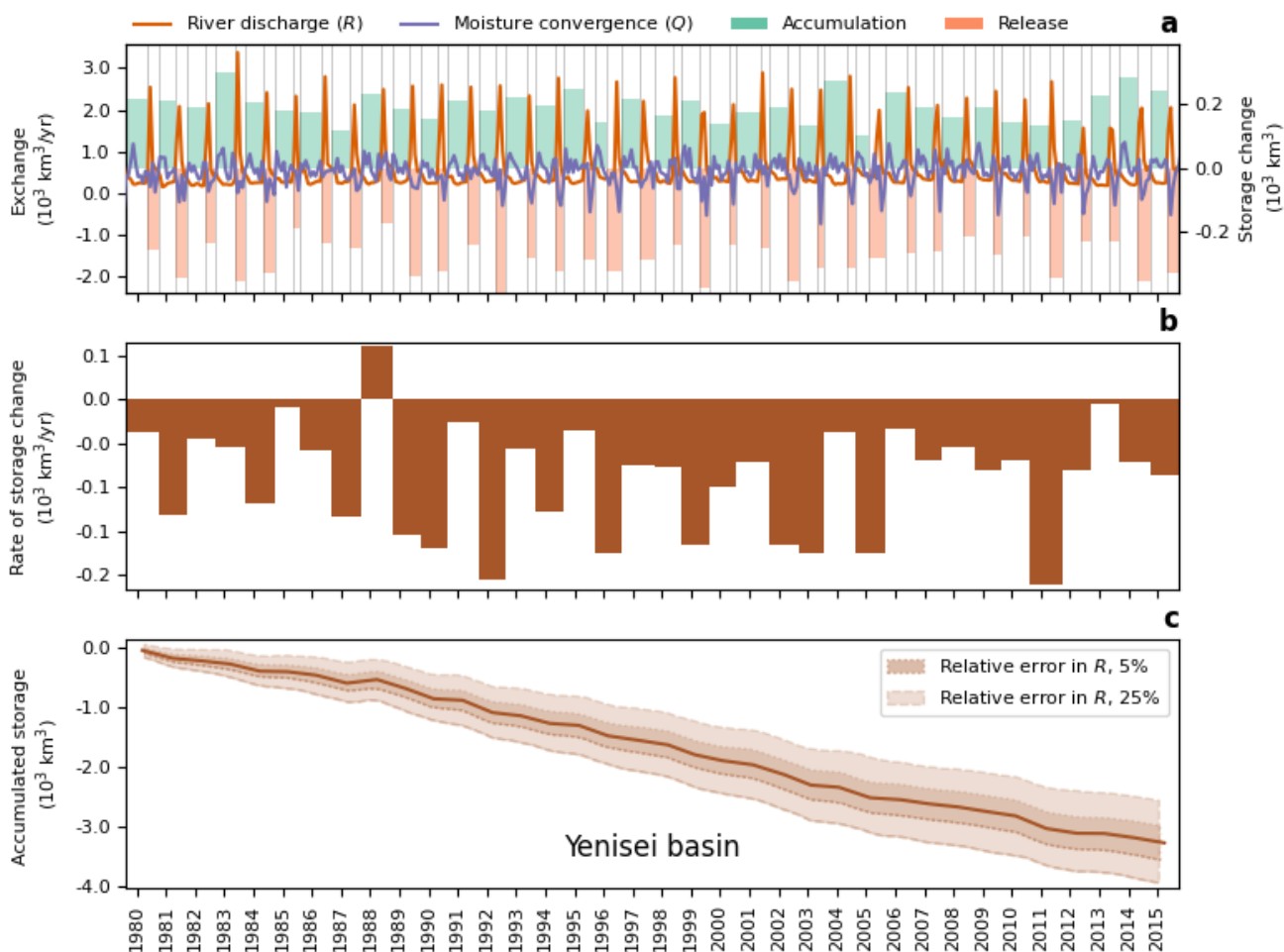

**Figure A25. LAR's dynamics in the Yenisei river basin. a,** Monthly river discharge $R$ and net atmospheric convergence $Q$ (left axis). Green and orange bars show, respectively, the extent and volume (right axis) of accumulation and release periods. **b,** Net change in the LAR water storage after pairs of consecutive storage and release periods. **c,** Cumulative change in the LAR's water storage, including the corresponding errors in convergence and estimated discharge uncertainties (shadowed bands). See Methods for more details.





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
