# Peer review of "Wetting and drying trends in the Land-Atmosphere Reservoir of large basins around the world"

_Hydrology and Earth System Sciences, 2023_

## Author Comment (AC1)

**Discussion of "Wetting and drying trends in the Land-Atmosphere Reservoir of large basins around the world" (manuscript number: hess-2023-172) — Reviewer 1**

Juan F. Salazar[1], Ruben D. Molina[1], Jorge I. Zuluaga[2], and Jesus D. Gomez-Velez[3]

[1]GIGA, Escuela Ambiental, Facultad de Ingeniería, Universidad de Antioquia, Calle 70 No. 52-21, Medellín, Colombia.
[2]SEAP/FACom, Instituto de Física, Facultad de Ciencias Exactas y Naturales, Universidad de Antioquia, Calle 70 No. 52-21, Medellín, Colombia.
[3]Environmental Sciences Division & Climate Change Science Institute, Oak Ridge National Laboratory, 1 Bethel Valley Road, Oak Ridge, TN, 37830, USA

**Correspondence:** Juan F. Salazar (juan.salazar@udea.edu.co)

This document presents comments by Reviewer 1 (blue font) and our responses (black font).

**Comment R1-1**

Salazar et al. introduces the concept of the Land-Atmosphere Reservoir (LAR), which explicitly includes atmospheric processes as a critical component of the basin water budget. This is in contrast to traditional approaches that treat atmospheric processes as an external forcing to the basin. They have undertaken a rigorous analysis of obtained time-series data of monthly river discharge from the HYdro-geochemistry of the AMazonian Basin (HYBAM) and the Global Runoff Data Centre (GRDC). They selected specific gauging stations for each of the six large basins studied and used data from the ERA5 reanalysis for the years 1979–2020 to estimate moisture convergence for each basin. The authors find considerable latitudinal trends in water storage within the studied basins' LAR, with tropical basins getting wetter and temperate basins getting drier.

While the findings are intriguing and somewhat intuitive given the state-of-knowledge in climate science, the paper could go deeper into the mechanisms (atmospheric and hydrologic processes) driving these latitudinal trends. I will elaborate on this below.

Thank you. We greatly appreciate your constructive comments and suggestions. Please find our responses below and note that they include new results that shed light on the mechanisms behind the trends and their latitudinal contrast.

**Comment R1-2**

**Challenges and questions in applying the LAR Framework to catchment hydrology**

The state-of-knowledge in catchment hydrology suggests that hydrologic processes (infiltration, flowpaths, runoff generation, etc.) are not too different in "tropical" (low-latitude, to be more precise) and "temperate" (mid- to high-latitude) catchments. Admittedly, most of these catchment hydrology studies are at spatial scales much smaller than the large basins consid-

20    ered by Salazar et al. And that is because of the obvious reason that understanding hillslope- and catchment-scale surface and subsurface processes is logistically limiting (if not impossible to perform) in large basins. That said, from the perspective of catchment hydrology (what the authors here refer to as "Land Reservoir"), the hydrologic processes in low and mid-latitude catchments are not too different. This then, in my view, begs the question: **what explains the latitudinal trends, from a process-based, mechanistic perspective, in both Land and Atmosphere domains?**

25    For reasons like the ones you presented, we do not think the latitudinal contrast in the trends arises from specific differences in hydrological processes between low- and high-latitude basins. We hypothesize that such contrast is caused mainly by land-atmosphere exchanges and atmospheric processes currently affected by climate change. Compared to high-latitudes, the low-latitude atmosphere is thicker and wetter, and its warming due to climate change increases its capacity to hold water. This is consistent with an increased capacity of the low-latitude LAR to store water.

30    High-latitude basins are warming, too, due to climate change. However, in such basins, the increased capacity of the atmosphere to hold water does not compensate for surface water losses due to snow and ice melting, leading to glaciers retreat and permafrost thawing. We hypothesize that high-latitude basins are losing more water due to these surface processes than they can gain due to atmospheric warming.

    Low-latitude glaciers are also retreating —they tend to disappear—, but they are concentrated in high-altitude mountains,

35  and their size is too small to govern the storage dynamics in large basins like the Amazon, Congo, and Parana. In contrast, snow and ice dynamics are much more significant in high-latitude basins.

    Motivated by your comment (and a similar Reviewer 2's comment), we will include the following two figures into the revised manuscript. They show, for each of the studied basins, a comparison between $d(S_L + S_A)/dt$ based on our Equation (2) and $dS_L/dt$ estimated from two different GRACE products. These new figures show three ideas we want to highlight. First, there

40  is a high correlation between the LAR storage change estimated with our Equation (2) and the LR storage change obtained from GRACE. Although the LAR and LR storages are not the same, they are related, and therefore, this correlation between time series obtained from substantially different sources helps validate our results.

    Second, there are two types of basins, as illustrated in Figure 1. In a basin like the Amazon, storage variations in the LAR are wider in amplitude than the corresponding variations in the LR. In contrast, in the Ob basin, LAR storage variations

45  largely coincide with LR storage variations. Our interpretation is that, in the first type of basins, land-atmosphere exchanges and atmospheric processes play a more prominent role in the storage dynamics than in the second type, where TWS largely controls these dynamics.

    Third, low-latitude basins pertain to the first type, whereas high-latitude basins are closer to the second type. This lends additional support to our hypothesis about the latitudinal contrast in the trends because, from this perspective, low-latitude

50  basins seem more sensitive to atmospheric changes (e.g., warming due to climate change) than high-latitude basins that are more sensitive to changes in terrestrial water (e.g., snow and ice loss).

    The revised manuscript will include these new results and discussion, as well as your suggestion of using "low-latitude" and "high-latitude" instead of "tropical" and "temperate." We avoided using "mid- to high-" to simplify the writing.

[Figure]

**Figure 1. Comparison between the storage dynamics in the LAR and LR**. **a,b** Rate of storage change in the LAR $(\mathrm{d}(S_A + S_L)/\mathrm{d}t)$ from Equation (2), and the corresponding estimates for the LR $(\mathrm{d}S_L/\mathrm{d}t)$ based on two different GRACE products: GRACE University of Texas and GRACE GSFC, for the Amazon and Ob basins. **c,d** Scatter plot, and **e,f** cross-correlation for different time lags between the LAR and LR time series.

**Comment R1-3**

The artificial reservoir analogy that the authors used to explain their results is simple, but too simplistic. It falls short in providing and explaining the underlying mechanisms. For the LAR framework to gain traction (as it should), it would have to be clear to catchment hydrologists how it could potentially change their worldview, i.e., how they perform their work both from experimental and modeling standpoints. I suggest that the authors address this comment by discussing the challenges and questions in applying LAR to catchment hydrology. Here's an outline or guide that they might find useful in the revision:

Thank you. We greatly appreciate your comment that the LAR framework should gain traction, as well the effort and time you invested not only in commenting but also in suggesting an outline. Below, we follow the suggested outline to respond.

**Comment R1-4**

1. Scale Mismatch: Catchment hydrology studies are often conducted at much smaller spatial scales than the large basins considered in the Land-Atmosphere Reservoir (LAR) framework. How can the LAR framework be adapted or scaled to be

[Figure]

**Figure 2.** Same as Fig. 1, but for the Parana, Congo, Mississippi, and Yenisei basins.

65   relevant for catchment hydrologists? Note that LMR is not exclusive to "large basins". LMR has also be found in small
     catchments, for example, in the low latitudes.

As you rightly point out, in principle, the LAR dynamics can be studied at any scale (i.e., for any basin size). No theoretical limitation exists, including that, as you said, Local Moisture Recycling (LMR) can occur at any basin.

However, whereas the LAR is crucial for understanding large basins, it might be unnecessary for small basins where external factors (e.g., large-scale wind patterns) largely impose precipitation. If so, LMR is possibly negligible, and therefore, the traditional LR framework is a parsimonious representation that works well without the complications of including the atmosphere in the control volume for the water budget computations.

That is why we focused on the largest basins on Earth, where LMR involves water amounts comparable in magnitude to other fluxes in the basin's water budget. Table 1 in the submitted manuscript shows that, for the studied basins, LMR represents between 23% and 47% of precipitation, which is comparable to evapotranspiration and river discharge in the same basins.

In contrast, we do not expect that LMR represents such a significant fraction of precipitation in small basins. This means that using the LAR for studying small basins should not produce significantly different results than the traditional LR. Hence, the LAR is crucial for studying large basins but not strictly necessary for small ones. What the limiting scale is is an intriguing question for future research.

Finally, studying small basins through the LAR lens is limited by the availability of atmospheric convergence estimates at the same scale. One could obtain these estimates with high-resolution atmospheric models, but they are not widely available, such as reanalysis data for large basins.

The revised manuscript will include this discussion about the applicability of the LAR framework at different scales.

**Comment R1-5**

[...] 2. Complexity vs. Simplicity: The LAR framework is simple, but it may be too simplistic to capture the nuances of hydrologic processes at the catchment level. How can the framework be refined to include more complex processes?

We agree that the LAR framework is simple. Indeed, it is based on computing the water budget at a given control volume: the LAR. Also, we understand your concern about whether it is too simplistic but prefer thinking that it is parsimonious and aligned with the recent evolution of hydrology to Earth system science, in the sense discussed by Sivapalan (2018, 2006). These discussions and perspectives, also presented by McDonnell et al. (2007), among other publications of the same group, were a crucial inspiration for our study, which we will better acknowledge in the revised manuscript.

Three key ideas of such perspectives about the evolution of hydrology, which we applied in our development of the LAR framework, are: first, that river basins are complex systems with emergent properties and patterns that can be observable despite the inherent complexity and heterogeneity behind them. For instance, river discharge exhibits interesting, observable patterns despite its dependence on myriad processes, including not only physical but also biological and social phenomena occurring in the river basin.

Second, storage and release of water are two basic functions of any river basin, which depend on a combination of processes that can be only partially disentangled (e.g., by simplifying them in a model) but somehow summarize that complexity and heterogeneity and produce large scale patterns such as the LAR trends. Thinking about these two basic functions is our motivation for presenting the reservoir analogy.

Third, we can learn about basins from a "top-down approach" that considers, in the first place, the large-scale patterns (e.g., a trend in the LAR) and then advance towards understanding the processes behind them. Your comment invites us, precisely, to advance in this direction. That is why, in our previous response, we presented the hypothesis of different climate change effects possibly explaining the latitudinal contrast in the trends.

Going deeper could be done in two ways. One way is to develop new models that use the LAR as a starting point for defining the control volume. This simple step leads to substantial changes relative to hydrological models developed with the LR as the control volume. For instance, whereas precipitation is an external input in the latter, it is an internal flow in the former.

Another way is to conduct basin-specific studies to explore the reasons for the LAR trends further. For instance, Reviewer 2 asked us about the reasons for the change in the trend in the Congo River basin. This is an interesting question we would like to answer. However, we have not found a sound hypothesis for this trend change so far, but we are confident that the data shows that. The explanation needs a more specific study of this basin.

However, developing such models or conducting such basin-specific studies is beyond the scope of our current study. We are sorry for mentioning this here; we are aware of your comment about this argument, but we sincerely believe such advances require new studies.

**Comment R1-6**

[...] 3. Interdisciplinary Integration: The LAR framework needs to be clear and compelling to catchment hydrologists to change (or refine!) their worldview. How can the framework be presented in a way that is relevant to multiple disciplines, including catchment hydrology?

We agree with you and hope the LAR framework will be relevant for different disciplines interested in rivers and basins. The revised manuscript will discuss further the following ideas:

- Water storage dynamics in large basins is critical for the sustainability of terrestrial ecosystems and societies. This is not a land- but a land-atmosphere dynamics, as incorporated into the LAR framework.

- The LAR trends should be monitored and discussed as a possible manifestation of climate change.

- Catchment hydrologists should consider whether the traditional LR framework is enough for a specific study or whether the LAR is needed. In principle, the LAR is needed for large basins with powerful LMR, which are still often studied using the traditional LR.

- The LAR highlights the importance of LMR for the water budget of large basins, highlighting the link between LULC change, including tropical deforestation, river discharge, and, more broadly, water security (e.g., Posada-Marín and Salazar, 2022).

130    – Linking river discharge to LMR through the LAR contributes to current debates about the hydrological role of forests and deforestation impacts, e.g., the contrast between the supply- and demand-side thinking described by Ellison et al. (2011) and the biotic pump concept debate (Makarieva and Gorshkov, 2007).

**Comment R1-7**

I would discourage the authors from resorting to the usual 'beyond the scope of this study' excuse that comes with [most]

135    revisions.

We hope we have provided specific responses to your comments despite using the "beyond the scope" argument once.

**Comment R1-8**

Other comment:

L294-295. In the previous page (L278), you suggested that TWS estimates from GRACE are contradictory. But here,

140    GRACE suddenly figures prominently and is used to support the claim that wetting and drying trends are "underway world-wide"? You cannot question GRACE's TWS estimates only to use the same to support your claim. That is a self-contradiction.

We intend not to say that GRACE estimates are invalid but to highlight that they have a level of uncertainty that allows contradictory results from different products for the same basin. This contradiction does not mean invalidity but set uncertainty levels. An analogy is the interpretation of different climate models predicting contradictory results for the same region, e.g.,

145    total precipitation increase versus decrease. Despite its uncertainties, GRACE provides robust evidence that Terrestrial Water Storage (TWS) is changing worldwide and that these changes have regional variations.

We will clarify this in the revised manuscript.

**References**

Ellison, D., Futter, M. N., and Bishop, K.: On the forest cover-water yield debate: From demand- to supply-side thinking, Global Change
Biology, 18, 806–820, https://doi.org/10.1111/j.1365-2486.2011.02589.x, 2011.

Makarieva, A. M. and Gorshkov, V. G.: Biotic pump of atmospheric moisture as driver of the hydrological cycle on land, Hydrology and
Earth System Sciences, 11, 1013–1033, https://doi.org/10.5194/hess-11-1013-2007, 2007.

McDonnell, J., Sivapalan, M., Vaché, K., Dunn, S., Grant, G., Haggerty, R., Hinz, C., Hooper, R., Kirchner, J., Roderick, M., et al.: Moving
beyond heterogeneity and process complexity: A new vision for watershed hydrology, Water Resources Research, 43, https://doi.org/10.
1029/2006WR005467, 2007.

Posada-Marín, J. A. and Salazar, J. F.: River flow response to deforestation: Contrasting results from different models, Water Security, p.
100115, https://doi.org/10.1016/j.wasec.2022.100115, 2022.

Sivapalan, M.: Pattern, process and function: elements of a unified theory of hydrology at the catchment scale, Encyclopedia of hydrological
sciences, 2006.

Sivapalan, M.: From engineering hydrology to Earth system science: milestones in the transformation of hydrologic science, Hydrology and
Earth System Sciences, 22, 1665–1693, https://doi.org/10.5194/hess-22-1665-2018, 2018.

---

## Author Comment (AC2)

**Discussion of "Wetting and drying trends in the Land-Atmosphere Reservoir of large basins around the world" (manuscript number: hess-2023-172) —Reviewer 2**

Juan F. Salazar[1], Ruben D. Molina[1], Jorge I. Zuluaga[2], and Jesus D. Gomez-Velez[3]

[1]GIGA, Escuela Ambiental, Facultad de Ingeniería, Universidad de Antioquia, Calle 70 No. 52-21, Medellín, Colombia.
[2]SEAP/FACom, Instituto de Física, Facultad de Ciencias Exactas y Naturales, Universidad de Antioquia, Calle 70 No. 52-21, Medellín, Colombia.
[3]Environmental Sciences Division & Climate Change Science Institute, Oak Ridge National Laboratory, 1 Bethel Valley Road, Oak Ridge, TN, 37830, USA

**Correspondence:** Juan F. Salazar (juan.salazar@udea.edu.co)

This document presents comments by Reviewer 2 (blue font) and our responses (black font).

**Comment R2-1**

The authors introduce the concept of the Land-Atmosphere Reservoir (LAR), which explicitly considers land-atmosphere interactions such as moisture recycling when computing a basin water budget. The LAR is in contrast to traditional approaches that assume atmospheric processes as external effects. Based on the LAR concept, the authors study long-term storage trends of the six largest river basins using river discharge data from HYBAM and GRDC and meteorological data from ERA5 reanalysis, and find a contrasting latitudinal trend, with tropical basins getting wetter and temperate basins getting drier. The study is interesting, and the topic is suitable for publication in the Hydrology and Earth System Sciences. However, I have some comments that should be addressed before publication.

Thank you. We greatly appreciate your constructive comments and suggestions. Please find our responses below and note that they include new results that shed light on the mechanisms behind the trends and their latitudinal contrast.

**Comment R2-2**

Is there any reason why the authors apply the LAR only to the largest basins? Given that GRDC and ERA5 data are available globally, a similar analysis could be conducted for other basins (with different sizes and climatic conditions) with relatively little effort.

In principle, the LAR dynamics can be studied at any scale (i.e., for any basin size). No theoretical limitation exists, including that Local Moisture Recycling (LMR) can occur at any basin.

However, there are theoretical and practical reasons for focusing on the largest basins. Whereas the LAR is crucial for understanding large basins, it might be unnecessary for small basins where external factors (e.g., large-scale wind patterns)

20  largely impose precipitation. If so, LMR is possibly negligible, and therefore, the traditional LR framework is a parsimonious representation that works well without the complications of including the atmosphere in the control volume for the water budget computations.

That is why we focused on the largest basins on Earth, where LMR involves water amounts comparable in magnitude to other fluxes in the basin's water budget. Table 1 in the submitted manuscript shows that, for the studied basins, LMR represents

25  between 23% and 47% of precipitation, which is comparable to evapotranspiration and river discharge in the same basins.

In contrast, we do not expect that LMR represents such a significant fraction of precipitation in small basins. This means that using the LAR for studying small basins should not produce significantly different results than the traditional LR. Hence, the LAR is crucial for studying large basins but not strictly necessary for small ones. What the limiting scale is is an intriguing question for future research.

30  Finally, studying small basins through the LAR lens is limited by the availability of atmospheric convergence estimates at the same scale. One could obtain these estimates with high-resolution atmospheric models, but they are not widely available, such as reanalysis data for large basins.

The revised manuscript will include this discussion about the applicability of the LAR framework at different scales. In future studies, we plan to use the LAR framework for more basins and look forward to other scientists doing that, too.

35  **Comment R2-3**

The authors applied the LAR concept to show the long-term trends in the large basins (e.g. Figs. 3 and 4). Is there something we can learn here that we didn't know from previous studies using the traditional approaches? I would like to see more detailed analysis and discussion in this aspect.

One of our study's key ideas is that applying the LAR framework to study large basins can yield substantially different

40  results than the traditional LR approach. In other words, the LAR allows us to learn lessons that would not be possible by using the LR, e.g., by modeling large basins from the traditional perspective of catchment hydrology. For instance, the trends in water storage for the LAR are related to but not equivalent to trends in TWS. This allows results such as the one for the Amazon basin, where the LAR trend exceeds the trends in TWS obtained from GRACE data by around one order of magnitude.

The main reason for these differences between the LAR and LR approaches is that the latter does not include LMR as

45  an internal mechanism of a complex basin system, which is critical in some basins like the Amazon where around 30% of precipitation is internally (i.e., within the LAR) recycled.

Our following response includes new results comparing TWS anomalies from GRACE and our estimates of water storage change in the LAR. This comparison shed light on the relationship between the LAR and LR. They are related but are not the same, especially in large basins where LMR plays a prominent role. Please continue this discussion in our subsequent response.

50  **Comment R2-4**

The paper has no in-depth explanation about physical mechanisms behind the revealed long-term trends of the basins. For instance, why do we see the contrasting wetting and drying trends between the tropical and temperate basins? Why has the trend in the Congo basin changed since 2000?

We hypothesize that the latitudinal contrast in the trends is caused mainly by land-atmosphere exchanges and atmospheric processes currently affected by climate change. Compared to high-latitudes, the low-latitude atmosphere is thicker and wetter, and its warming due to climate change increases its capacity to hold water. This is consistent with an increased capacity of the low-latitude LAR to store water.

Before continuing, please note that we are using "low-latitude" and "high-latitude" basins instead of "tropical" and "temperate" basins, following a good suggestion by Reviewer 1.

High-latitude basins are warming, too, due to climate change. However, in such basins, the increased capacity of the atmosphere to hold water does not compensate for surface water losses due to snow and ice melting, leading to glaciers retreat and permafrost thawing. We hypothesize that high-latitude basins are losing more water due to these surface processes than they can gain due to atmospheric warming.

Low-latitude glaciers are also retreating —they tend to disappear—, but they are concentrated in high-altitude mountains, and their size is too small to govern the storage dynamics in large basins like the Amazon, Congo, and Parana. In contrast, snow and ice dynamics are much more significant in high-latitude basins.

Motivated by your comment (and a similar Reviewer 1's comment), we will include the following two figures into the revised manuscript. They show, for each of the studied basins, a comparison between $d(S_L + S_A)/dt$ based on our Equation (2) and $dS_L/dt$ estimated from two different GRACE products. These new figures show three ideas we want to highlight. First, there is a high correlation between the LAR storage change estimated with our Equation (2) and the LR storage change obtained from GRACE. Although the LAR and LR storages are not the same, they are related, and therefore, this correlation between time series obtained from substantially different sources helps validate our results.

Second, there are two types of basins, as illustrated in Figure 1. In a basin like the Amazon, storage variations in the LAR are wider in amplitude than the corresponding variations in the LR. In contrast, in the Ob basin, LAR storage variations largely coincide with LR storage variations. Our interpretation is that, in the first type of basins, land-atmosphere exchanges and atmospheric processes play a more prominent role in the storage dynamics than in the second type, where TWS largely controls these dynamics.

Third, low-latitude basins pertain to the first type, whereas high-latitude basins are closer to the second type. This lends additional support to our hypothesis about the latitudinal contrast in the trends because, from this perspective, low-latitude basins seem more sensitive to atmospheric changes (e.g., warming due to climate change) than high-latitude basins that are more sensitive to changes in terrestrial water (e.g., snow and ice loss).

So far, we have not found a sound hypothesis for the trend change in the Congo River basin. The data shows such a change, but the explanation needs a more specific study of this basin, which might motivate future research.

The revised manuscript will include these new results and discussion.

[Figure]

**Figure 1. Comparison between the storage dynamics in the LAR and LR**. **a,b** Rate of storage change in the LAR ($d(S_A + S_L)/dt$) from Equation (2), and the corresponding estimates for the LR ($dS_L/dt$) based on two different GRACE products: GRACE University of Texas and GRACE GSFC, for the Amazon and Ob basins. **c,d** Scatter plot, and **e,f** cross-correlation for different time lags between the LAR and LR time series.

**Comment R2-5**

*Just as a minor suggestion, Fig. A9 to 14 and Fig. A15 to 20 can be combined into a figure, respectively, to avoid too many figures.*

Thank you, the revised manuscript will combine these figures as suggested.

[Figure]

**Figure 2.** Same as Fig. 1, but for the Parana, Congo, Mississippi, and Yenisei basins.